

# Environmental drivers of coccolithophore abundance and calcification across Drake Passage (Southern Ocean)

Anastasia Charalampopoulou[1], Alex J. Poulton[2], Dorothee C. E. Bakker[3], Mike I. Lucas[4], Mark C. Stinchcombe[2], and Toby Tyrrell[1].

[1] Ocean and Earth Science, National Oceanography Centre Southampton, University of Southampton, SO14 3ZH, UK.

[2] Ocean Biogeochemistry and Ecosystems, National Oceanography Centre, Southampton, SO14 3ZH, UK.

[3] School of Environmental Sciences, University of East Anglia, Norwich Research Park, NR4 7TJ, UK.

[4] Marine Research Institute, University of Cape Town, Rondebosch, South Africa.

*Correspondence to:* A. J. Poulton (Alex.Poulton@noc.ac.uk)

**Abstract.** Although coccolithophores are not as common in the Southern Ocean as they are in sub-polar waters of the North Atlantic, a few species, such as *Emiliania huxleyi*, are found during the summer months. Little is actually known about the calcite production (CP) of these communities, or how their distribution and physiology relates to environmental variables in this region. In February 2009, we made observations across Drake Passage (between South America and the Antarctic Peninsula) of coccolithophore distribution, CP, primary production, chlorophyll-*a* and macronutrient concentrations, irradiance and carbonate chemistry. Although CP represented less than 1% of total carbon fixation, coccolithophores were widespread across Drake Passage. The B/C morphotype of *E. huxleyi* was the dominant coccolithophore, with low estimates of coccolith calcite (~0.01 pmol C coccolith$^{-1}$) from biometric measurements. Both cell-normalised calcification (0.01-0.16 pmol C cell$^{-1}$ d$^{-1}$) and total CP (<20 μmol C m$^{-3}$ d$^{-1}$) were much lower than those observed in the sub-polar North Atlantic where *E. huxleyi* morphotype A is dominant. However, estimates of coccolith production rates were similar (0.1-1.2 coccoliths cell$^{-1}$ h$^{-1}$) to previous measurements made in the sub-polar North Atlantic. A multivariate statistical approach found that temperature and irradiance together were best able to explain the observed variation in species distribution and abundance (Spearman's rank correlation ρ = 0.4, p<0.01). Rates of calcification per cell and coccolith production, as well as community CP and *E. huxleyi* abundance, were all positively correlated (p<0.05) to the strong latitudinal gradient in temperature, irradiance and calcite saturation states across Drake Passage. Broadly, our results lend support to recent suggestions that coccolithophores, especially *E. huxleyi*, are



advancing pole-wards. However, our *in situ* observations indicate that this may owe more to sea-surface warming and increasing irradiance rather than increasing $CO_2$ concentrations.

## 1  Introduction

As major pelagic calcifiers, coccolithophores have received significant interest over the last couple of decades, mainly due to their importance in the marine carbon cycle where they contribute to both the organic carbon (biological) and carbonate pumps. *Emiliania huxleyi*, the most widespread coccolithophore species, has been extensively studied in laboratory cultures in terms of its physiology (e.g., Paasche, 2002), and there are now a growing number of field studies examining its eco-

physiology (e.g., Harley et al., 2010; Poulton et al., 2010, 2011, 2013, 2014; Charalampopoulou et al., 2011; Young et al., 2014).

There are currently several distinct morphotypes of *E. huxleyi* recognised in the literature, each exhibiting slightly different characteristics in terms of their coccoliths; the main morphotypes are currently termed A, B, B/C and R (Young et al., 2003, 2014; Poulton et al., 2011). Those most

commonly available in laboratory culture are morphotypes A and B (Paasche, 2002), although strains of B/C are now being isolated and studied from the Southern Ocean (e.g., Cook et al., 2011, 2013). Morphotype A forms blooms in the sub-polar North Atlantic and Norwegian coastal waters, whereas morphotype B is primarily found in the North Sea (van Bleijswijk et al., 1991; Holligan et al., 1993; Paasche et al., 1996). The eco-physiology of these two morphotypes is thus relatively well studied and

seasonally shallow mixed layers, high temperatures and high irradiances are often associated with their large-scale mono-specific blooms (Tyrrell and Merico, 2004; Raitsos et al., 2006).

Morphotype B/C is the dominant morphotype of *E. huxleyi* found in the colder waters of the Southern Ocean (Cubillos et al., 2007; Cook et al., 2011, 2013; Poulton et al., 2011, 2013). A number of observations have shown that *E. huxleyi* B/C is widespread in the Atlantic, Indian and Pacific sectors

of the Southern Ocean, except close to the Antarctic continent where coccolithophores are absent (Cubillos et al., 2007; Findlay and Giraudeau, 2000; Gravalosa et al., 2008; Mohan et al., 2008; Hinz et al., 2012). In contrast with the well-studied morphotypes A and B, most of the current knowledge on morphotype B/C regards the different morphological characteristics of its coccoliths, which have a lower calcite content relative to their northern hemisphere counterparts (Cook et al., 2011; Poulton et

al., 2011). Furthermore, the B/C morphotype is physiologically and genetically distinct from morphotype A (Cook et al., 2011, 2013), and in 2008 it dominated a Patagonian Shelf bloom which occurred in waters with low temperatures ($<8^{o}C$), high macronutrient concentrations (e.g., nitrate $>10$ $\mu mol\ kg^{-1}$) and high irradiances (Poulton et al., 2011, 2013; Balch et al., 2014). This dominance has now been suggested to result in low integrated calcite concentrations within bloom waters on the

Patagonian Shelf (Poulton et al., 2013; Balch et al., 2014).





Laboratory and field studies on *E. huxleyi* show that calcification in this species depends strongly on irradiance and is also stimulated by nutrient stress (Paasche, 2002; Zondervan, 2007; Müller et al., 2008), which give important clues to its response to changing nutrient and light conditions in the future ocean. However, the response of this species to increased $pCO_2$ (and associated reduced pH, i.e.

ocean acidification) in terms of growth and calcification rates appears closely related to the strain (both within and between morphotypes) studied (Langer et al., 2009, 2011) and the length of exposure (Lohbeck et al., 2012). Moreover, the sensitivity of *E. huxleyi* to elevated $pCO_2$ also depends on irradiance (Feng et al., 2008; Zondervan et al., 2002) and nutrient conditions (Sciandra et al., 2003; Delille et al., 2005). Future oceanic changes are likely to happen simultaneously (Gruber, 2011), due

to sea-surface warming (Barnett et al., 2005), shallowing of the mixed layer (Levitus et al., 2000) and ocean acidification (Orr et al., 2005). Thus, it is important to validate the findings of laboratory experiments with field studies in order to understand how natural coccolithophore populations might respond to simultaneous changes in environmental variables.

In natural coccolithophore populations, bulk community rates of calcite production (CP) are

influenced by both cell numbers (a product of growth and mortality of the population), and by variability in the rate of calcification per cell (a function of environmental conditions and species composition) (Poulton et al., 2010, 2014; Charalampopoulou et al., 2011). Cellular rates of calcification are dependent on both the calcite content of individual coccoliths and the rate of coccolith production. Rather than examining bulk CP and comparing it to environmental factors, a more

appropriate approach is to consider cellular levels of calcification in the form of cell-normalised rates (cell-CF), which when normalised to coccolith calcite content allow an estimate of coccolith production rates (Poulton et al., 2010, 2013; Charalampopoulou et al., 2011). The calcite content of individual coccoliths is determined by their volume/shape and distal shield length (DSL) (Young and Ziveri, 2000). Hence, with information on the species (or morphotype) present and measurements of

coccolith size it is possible to estimate coccolith calcite content (e.g., Poulton et al., 2011; Young et al., 2014; Jin et al., in review). Together, these three metrics (cell-CF, coccolith calcite, coccolith production rates) allow for greater insight into the calcification of natural communities than can be obtained from bulk CP measurements alone.

The Southern Ocean has naturally low calcite saturation states ($\Omega_C$), due to low seawater temperatures,

and will be among the first oceanic regions to experience widespread $CaCO_3$ under-saturation at the surface (Hauri et al., 2015). Nevertheless, there are indications that the distribution of *E. huxleyi* has recently extended pole-wards in the Southern Ocean (Winter et al., 2014), with a north to south progression from high to low coccolith-calcite containing morphotypes as temperature and $\Omega_C$ decline (Cubillos et al., 2007), often regarded as indicating a reduction in calcification rates in Antarctic



waters. In this context, the lack of information on coccolithophore calcification in the Southern Ocean
is a significant gap in our understanding of the effects of future changes on the marine carbon cycle.

The main aims of this study were to investigate the distribution and calcification of coccolithophore
populations across Drake Passage in the Southern Ocean, and to examine how these relate to
environmental variables (i.e., temperature, salinity, nutrients, irradiance, and carbonate chemistry
parameters). In order to examine changes in calcification, we made estimates of cellular coccolith
production rates (Poulton et al., 2010, 2013; Charalampopoulou et al., 2011) from cell-normalized
calcification rates (cell-CF) and estimates of coccolith calcite content using biometric measurements
(Poulton et al., 2011; Young et al., 2014; Jin et al., in review). Examination of such trends in cellular
calcification across environmental gradients allows for changes due to variability in both cell numbers
and cellular calcification to be taken into account: bulk measurements of (community) CP may be
driven by variability in both (Poulton et al., 2010, 2014).

## 2        Methods

### 2.1        Sampling

Sampling was conducted during cruise JC031 (03/02/2009 - 03/03/2009) on board the *RRS James*
*Cook*, from Punta Arenas, Chile, to the Antarctic Peninsula (Transect 1) and back to the Falklands
(Transect 2) (Fig. 1). A stainless steel Conductivity-Temperature-Depth (CTD) rosette was deployed
at every sampling station and samples were collected from the upper 100 m of the water column.
Water samples for macronutrients, chlorophyll-*a* (Chl-*a*), and ancillary parameters (temperature,
salinity, irradiance) were collected at a total of 61 stations. Samples for coccolithophore abundance
and diversity were collected at 53 stations, carbonate chemistry parameters at 5 m depth were
measured at 51 stations, and samples for CP and primary production were collected at 20 stations (red
circles in Fig. 1).

### 2.2        Study area

Drake Passage is characterized by strong eastward flow of the Antarctic Circumpolar Current (ACC)
driven by strong westerly winds. The boundaries of the ACC are defined by oceanic fronts, where
rapid changes in temperature and salinity occur over short distances. The northern boundary of the
ACC is the Subtropical Front (STF), which separates warm sub-tropical waters from cold sub-
Antarctic waters (Orsi et al., 1995; Pollard et al., 2002). South of the STF, the three fronts associated
with the ACC are, from north to south: the Sub-Antarctic Front (SAF), the Polar Front (PF) and the
Southern ACC Front (SACCF). These three fronts define the three major zones of the ACC (Fig. 1).
The area between the STF and the SAF is often referred to as the Sub-Antarctic Zone, while between
the SAF and the PF is the Polar Frontal Zone and between the PF and the SACCF is the Antarctic





Zone (Orsi et al., 1995; Pollard et al., 2002). A fourth zone, located between the southern boundary of the ACC (SB) and the Antarctic continent, is referred to as the Continental Zone (Whitworth, 1980).

The positions of these fronts during the two transects of the cruise are shown in Figure 1. Dynamic height (Sun and Watts 2001), as well as the criteria of Orsi et al. (1995), were used to determine the positions of the fronts (S. Close and G. Evans, unpublished data 2009)

### 2.3 Coccolithophore community

Water samples (1 L) from up to 5 CTD depths over the upper 100 m were gently filtered onto

Millipore Isopore membrane filters (25 mm diameter, 1.2 $\mu$m pore size), with a 25 mm diameter circle of 10 $\mu$m nylon mesh acting as a backing filter to achieve even distribution of cells. The membrane filters were rinsed with trace ammonia solution (pH 9 - 10) to remove salts, oven dried overnight at 30ºC and stored in the dark in sealed Petri dishes. A radially cut portion of each filter was mounted on an aluminium stub and gold-coated. For each filter, 225 fields of view (FOV, = images), together

covering an area of ~ 0.9 mm$^2$ of the filter paper, were taken at 5000X magnification along a predefined meander-shaped transect, using a Scanning Electron Microscope (Leo 1450VP, Carl Zeiss, Germany) combined with SmartSEM software. For each sample, both complete coccospheres and detached coccoliths were enumerated until 300 of each were reached or until all FOV had been counted. The SmartSEM software was set to scan for zero overlap between FOVs. To avoid double-

counting of specimens that were on the FOV margins, only cells at the top and right edges of each FOV were counted but not at the bottom and left edges. The number of FOVs counted was used to calculate the area of the filter covered (the size of one FOV was 4.054×10$^{-3}$ mm$^2$).

Both coccospheres and coccoliths were identified to species level following Young et al. (2003), and the abundance of these for each species was calculated as:

$$\text{Coccospheres or detached coccoliths mL}^{-1} = C \times (F/A) / V \qquad (1)$$

where C is the total number of coccospheres or coccoliths counted, A is the area investigated (mm$^2$), F is the total filter area (mm$^2$) and V is the volume filtered (mL). Standard error (S.E.) of the coccolithophore counts was calculated as the square root of the counted number of cells ($\sqrt{C}$) divided by the equivalent volume of sample investigated (A×V/ F) (Taylor, 1982). Ninety five per cent (95%)

confidence limits of coccolithophore abundance were obtained by multiplying the standard error by the appropriate z-score where more than 30 cells were counted, and so the uncertainty was $\pm$ 1.96 × S.E. cells mL$^{-1}$. When less than 30 cells were counted the appropriate t-values were used instead of z-scores (Fowler et al., 1998).





### 2.4 Coccolith morphology and calcite content

Detached coccoliths were predominantly (>99%) from the species *Emiliania huxleyi*. The distal shield lengths (DSL) of 50 *E. huxleyi* detached coccoliths were measured in each of the 20 stations where CP was also measured, using the image processing software ImageJ (Abramoff et al., 2004; Young et al., 2014). At the same time, each coccolith was classified as either morphotype A, B or B/C following Young et al. (2003) and Poulton et al. (2011). Only type B/C was found in our samples, as identified by its delicate distal shield elements and central area open or with a thin plate. Coccolith calcite content (pmol C or $CaCO_3$ coccolith$^{-1}$) was calculated from the volume of each coccolith and the density (2.7 pg µm$^{-3}$) and molecular weight (100 g mol$^{-1}$) of calcite. Coccolith volume was a function of DSL and a shape dependent constant, $k_s$ (0.015 for type B/C; Poulton et al., 2011) following the equations of Young and Ziveri (2000):

$$\text{Coccolith calcite (pmol C or } CaCO_3 \text{ coccolith}^{-1}) = k_s \times DSL^3 \times \text{calcite density/molecular weight} \quad (2)$$

### 2.5 Calcite production and primary production

Water samples for rate measurements were collected before dawn (or early in the morning in a few cases), from 3-5 light depths within the upper 100 m of the water column (including 0.1-1.5%, 7%, 14%, 33% and 55% of incident Photosynthetically Active Radiation, PAR) or only from surface waters (55%). Daily rates of primary production (PP) and calcite production (CP) were determined following the 'micro-diffusion' technique of Paasche and Brubak (1994), as modified by Balch et al. (2000). Water samples (each 150 mL, 3 replicates plus 1 formalin-killed blank) were collected from each light depth, spiked with 100 µCi of $^{14}$C-labelled sodium bicarbonate (Perkin Elmer, UK) and incubated in on-deck incubators for 24 hrs. Light depths were replicated using a combination of misty blue and neutral density (Lee filters, UK) filters, and samples were kept at ambient sea surface temperature by providing a continuous flow of water from the underway supply through the on-deck incubators.

Incubations were terminated by filtration through 25 mm 0.2 µm polycarbonate Isopore filters, which were then acidified with 1 mL of 1% phosphoric acid to separate the inorganic fraction (labile, CP) from the organic fraction (non-labile, PP). The inorganic fraction was captured as $^{14}$C-$CO_2$ on a ß-phenylethylamine soaked Whatman GF/A filter and placed in a separate liquid scintillation vial to the original filter. Liquid scintillation cocktail was added to both vials and activity was measured on a TriCarb liquid scintillation counter. Counts were converted to uptake rates using standard methods. The average relative standard deviation (calculated as SD $\times$ 100/mean) of triplicate measurements was 6% (1-19%) for PP and 36% (2-86%) for CP, with the higher deviations observed at the base of the euphotic zone were rates of PP and CP were close to zero. The formalin blanks represented a significant proportion of the CP signal (mean 35%, range 5-87%) from the upper 50 m. Blank



contribution was even higher below 75 m and close to Antarctica, where the CP rates were very low.

Similar high blank contributions have been reported in other studies (e.g., Poulton et al., 2013, 2014). In contrast, the blanks represented only ~1% of the PP signal.

### 2.6    Cell-specific calcification and coccolith production

Cell specific calcification (cell-CF) was calculated from total CP and coccolithophore abundance. The associated error was derived from propagation of the individual errors of total CP ($\pm$ 0-3.5 µmol C m$^{-3}$

d$^{-1}$) and coccolithophore abundance ($\pm$ 0-53 cells mL$^{-1}$) following Taylor (1982). Daily coccolith production per cell at each station was calculated from cell-CF and coccolith calcite estimated for each station. The uncertainty in coccolith production rates at each station, due to associated errors in measurements of CP and coccolithophore abundance, was smaller than the range of coccolith production rates observed due to the relatively high variation in coccolith size. Hence, differences in

coccolith production between stations could be attributed mainly to changes in coccolith size rather than to errors associated with the method by which these were estimated.

### 2.7    Environmental variables

To assess how coccolithophore abundance, species composition and cellular calcification compare to environmental variables, several sets of ancillary measurements were used. These included

temperature, salinity, macronutrient (phosphate, nitrate and silicic acid) concentrations, Chl-*a*, carbonate chemistry parameters, and levels of mixed layer irradiance. Temperature and salinity values were extracted from the Seabird 911+ CTD package. Phosphate, nitrate and silicic acid micro-molar concentrations were determined using a Scalar San Plus Auto-analyser following the methods described by Kirkwood (1996). The errors associated with phosphate, nitrate and silicic acid analysis

for this cruise were $\pm$ 0.01, $\pm$ 0.16 and $\pm$ 0.05 µmol kg$^{-1}$, respectively. Water samples (0.2-0.25 L) for chlorophyll-*a* (Chl-*a*) analysis were filtered onto Whatman GFF (~0.7 µm pore size) filters and extracted in 8 mL 90% acetone for 24 h in the dark at 4$^{o}$C. Chl-*a* fluorescence was measured on a Turner Designs AU-10 fluorometer equipped with Welschmeyer (1994) filters and calibrated using a pure Chl-*a* standard (Sigma).

Methodology for Dissolved Inorganic Carbon (C$_T$) and Total Alkalinity (A$_T$) sampling and analysis followed Bakker et al. (2009). Water samples were drawn into 500 mL Schott ® SUPRAX borosilicate glass bottles following Dickson et al. (2007) to minimize gas exchange. Samples were usually analyzed within 6 hours of collection; when such rapid analysis was not possible the samples were poisoned with 100 µL of a saturated solution of mercuric chloride (7 g per 100 mL). Three

different instruments were used for C$_T$ and A$_T$ analysis. The first was used for C$_T$ only and has an extractor unit built after the design by Robinson and Williams (1992), operating at 4ºC. The second was a VINDTA 3C combined C$_T$/T$_A$ instrument (Marianda, Germany) operating at 25°C. The third



was another VINDTA 3C, which was used for determining $A_T$ after $C_T$ analysis on the stand-alone $C_T$ extractor. Water samples were analyzed for $C_T$ by the coulometric method after Johnson et al. (1987).

The $A_T$ measurements were made by potentiometric titration with the two VINDTA 3C instruments. Two replicate analyses were made on each sample bottle, and replicate samples were also drawn from the CTD rosette. Certified Reference Materials (CRMs) (from A.G. Dickson, Scripps Institute of Oceanography) were used for instrument calibration and at least two CRMs were run per station. The precision and accuracy for both $A_T$ and $C_T$ was <3 μmol kg$^{-1}$. Calcite saturation state ($\Omega_C$), $pH_T$ (pH on

the total scale) and $pCO_2$ were calculated from DIC, TA, nutrients, temperature, salinity and pressure data using the CO2SYS.XLS program for 5 m depth (Pierrot et al., 2006).

To calculate average daily irradiances over the mixed layer, the mixed layer depth (MLD) was determined as the shallowest depth corresponding to a density difference ($\Delta\sigma_t$) with surface waters of more than $\Delta\sigma_t = 0.03$ sigma units, as has been recommended for the Southern Ocean (Dong et al.,

2008). The vertical attenuation coefficient ($k_d$) for PAR for downward irradiance at each station was calculated from the monthly averaged (February 2009) light attenuation coefficient at 490 nm wavelength ($k_{490}$) estimate by the MODIS ocean colour satellite (http://oceancolor.gsfc.nasa.gov/) following Rochford et al. (2001):

$$k_d = 0.0085 + 1.6243 \times k_{490} \qquad (3)$$

The relationship describing the exponential attenuation of downward irradiance ($E_z$) with depth (z) is:

$$E_z = E_0 \times \exp(-k_d \times z) \qquad (4)$$

where $E_0$ is the instantaneous subsurface irradiance. $E_0$ was calculated from minute averaged PAR above the sea surface ($PAR_{above\ surface}$) data, obtained from the ship-mounted sensors, assuming $E_0$ was 55% of $PAR_{above\ surface}$ (Kirk, 1983). Daily $PAR_{above\ surface}$ was calculated as the sum of the minute

averaged data over 24 hours and daily irradiance was then calculated at every 1 m down to the MLD:

$$E_{z,\ daily} = 0.55 \times daily\ PAR_{above\ surface} \times \exp(-k_d \times z) \qquad (5)$$

The average irradiance over the mixed layer, $E_{MLD}$ (mol PAR m$^{-2}$ d$^{-1}$), was then calculated as the sum of $E_{z,\ daily}$ at every 1 m down to the MLD, and then divided by the MLD. The euphotic zone depth at each station ($Z_{eu}$), defined as the depth at which $E_z$ falls to 1% of the subsurface value, was equal to an

optical depth of 4.6 and hence $Z_{eu} = 4.6/k_d$ (Kirk, 1983). Comparison of daily PAR data from the ship's sensor with a 32 d composite of MODIS PAR data during the study period showed good agreement and confirmed that daily PAR values were typical for the time of the year and were not biased by weather conditions.

### 2.8   Statistical analysis



Multivariate statistics were used to relate spatial changes in coccolithophore abundance, species

composition and cellular calcification to changes in environmental variables. This was carried out

following the methods described by Clarke (1993), using E-PRIMER (v. 6.0) (Clarke and Gorley,

2006). Analysis of biotic data was carried out on square-root-transformed ($\sqrt{x}$) species abundances,

using Bray-Curtis similarity to determine changes in the abundance of both dominant and less

abundant species. Analysis of abiotic data was carried out on power transformed (to reduce skewness

and stabilize the variance) and standardised (to bring all variables to comparable scales) values of

temperature, salinity, phosphate, nitrate, $pH_T$, $\Omega_C$, $E_{MLD}$, and $E_0$, using Euclidean distance to determine

spatial changes in these variables. Principal Component Analysis (PCA) was carried out on

environmental data to reduce the 8-fold variability to a low-dimensional representation of spatial

changes in these variables. The BEST routine was used to search for relationships between the biotic

and abiotic patterns and to identify which environmental variables(s) explained most of the variation

in coccolithophore distribution. Spearman's rank correlations were also used to identify relationships

between calcification parameters (coccolith calcite content, coccolith production rate, total and cell-

specific calcification) and environmental variables.

**3      Results**

**3.1      Environmental variables and chlorophyll**

Most physicochemical variables exhibited a strong north-south trend, with temperature and salinity

decreasing towards Antarctica (Fig. 2A). Temperature was highest to the south of Chile and the

Falkland Islands (8-9ºC) and lowest off the Antarctic Peninsula (~ 2ºC). Salinity only varied by 0.4,

with the highest values (~ 34.1) associated with the SAF on Transect 1, and the lowest values (~ 33.7-

33.8) observed in the Antarctic Zone just north of the SACCF on both transects. Macronutrient

concentrations increased towards Antarctica (Fig. 2B), with nitrate values between 16.5 and 27.0 μmol

$kg^{-1}$, phosphate values between 1.2 and 1.8 μmol $kg^{-1}$, and silicic acid showing rapid changes

associated with frontal positions and ranging from 1.4 to 50.9 μmol $kg^{-1}$. Such high nitrate and

phosphate concentrations are unlikely to be limiting and the nitrate to phosphate ratio ranged from

14:1 to 16:1. The silicic acid to nitrate ratio was 2:1 close to Antarctica but fell to 1:13 north of the PF,

suggesting silicic acid limitation north of the PF. $pH_T$ fluctuated by 0.08 units but did not exhibit a

clear latitudinal trend (Fig. 2C). The highest $pH_T$ value (8.12) was found at station 24, between the two

branches of the PF on Transect 1, and the lowest value (8.04) was calculated for station 63, just north

of the SACCF (N) on Transect 2. Calcite saturation states ($\Omega_C$) ranged between 2.5 and 3.3 and

exhibited an overall decrease towards Antarctica (Fig. 2C).

Euphotic zone depths ($Z_{eu}$, 39-100 m) were generally deeper than the MLD (15-65 m) across both

transects (Fig. 2D), indicating that phytoplankton were not likely to be mixed down to depths where



there was insufficient light. $Z_{eu}$ was deepest in the Antarctic Zone, just north of the SACCF on both

transects, and shallowest close to the continental shelves of Chile, Antarctica and the Falkland Islands. MLD did not exhibit a clear trend and was shallowest off the Chile and Falkland shelves and at a number of stations in the Sub-Antarctic and Antarctic Zones.

Surface Chl-*a* concentrations were generally low (<0.5 mg m$^{-3}$) across most of Drake Passage, with an average Chl-*a* of 0.26 mg m$^{-3}$ (Fig. 2D). Higher Chl-*a* concentrations were observed in the Sub-

Antarctic and Continental Zones, with the maximum (0.97 mg m$^{-3}$) associated with the SAF on Transect 2. The lowest Chl-*a* concentrations were observed in the Antarctic Zone, where surface concentrations were <0.3 mg m$^{-3}$.

Daily PAR above the sea surface (PAR$_{above\ surface}$) ranged from 12 to 46 mol PAR m$^{-2}$ d$^{-1}$ (Fig. 2E). The highest PAR$_{above\ surface}$ was observed close to the Chile and Falkland shelves and decreased sharply

towards the SAF. High values were also observed at some stations south of the PF and SAF. Low PAR$_{above\ surface}$ was observed close to Antarctica, especially on Transect 1. Average mixed layer irradiance ($\bar{E}_{MLD}$) ranged between 2 and 12 mol PAR m$^{-2}$ d$^{-1}$ and generally followed the PAR$_{above\ surface}$ distribution, with the exception of a few stations where MLD was exceptionally shallow resulting in relatively high $\bar{E}_{MLD}$ (Fig. 2E).

Principal Component Analysis (PCA) of environmental variables (Fig. 3, Table 1) showed that the first principal component (PC-1) explained 62.4% of the variation in environmental variables and PC-1 and PC-2 explained 80.6%. PC-1 was a linear combination of mainly temperature, phosphate, nitrate and $\Omega_C$, with PC-1 and nutrients being anti-correlated to temperature and $\Omega_C$ (Fig. 3, Table 1). PC-2 was the linear combination of mainly $\bar{E}_{MLD}$ and pH, with PC-2 and pH being anti-correlated to $\bar{E}_{MLD}$

(Fig. 3, Table 1).

### 3.2    Coccolithophores

Fifteen coccolithophore species were identified in the samples across Drake Passage: including *Emiliania huxleyi*, *Acanthoica quattrospina*, *Calciopappus caudatus*, *Calcidiscus leptoporus*, *Gephyrocapsa ericsonii*, *G. muellerae*, *G. ornata*, *Ophiaster hydroideus*, *Pappomonas* spp.,

*Papposphaera* spp., *Rhabdosphaera xiphos*, *Syracosphaera dilatata*, *S. halldalii*, *S. molischii*, and *Wigwamma antarctica* (Charalampopoulou, 2011). *Emiliania huxleyi* occurred all the way across Drake Passage, comprising an average of 89% of the total community in surface samples, with a range from 50% to 100% (Fig. 4). At only three stations did the relative contribution of *E. huxleyi* fall below 80% of total cell numbers, with two of these stations south of the Southern Boundary in the

Continental Zone (36 and 45).





Maximum coccolithophore abundance was associated with the SAF on both transects. Along Transect 1, *E. huxleyi* reached a maximum of ~580 cells mL$^{-1}$ in the Polar Frontal Zone and 289 cells mL$^{-1}$ between the two branches of the PF (Fig. 4). Along Transect 2, *E. huxleyi* abundance was generally <200 cells mL$^{-1}$, but peaked at 260 cells mL$^{-1}$ at the SAF. In the Continental Zone near to Antarctica, it

was only found in deeper samples (>25 m) at relatively low abundances (<13 cells mL$^{-1}$). *Gephyrocapsa muellerae* was characteristic of the Sub-Antarctic Zone south of the Falkland Islands (Fig. 4), occasionally contributing 10-36% towards total abundance. *Pappomonas* sp. and *W. antarctica* were found at low abundances in all regions and were the only coccolithophore species found in the surface waters of the Continental Zone. *Acanthoica quattrospina* and *C. caudatus* were

found in the Sub-Antarctic Zone of both transects and the Polar Frontal Zone on Transect 2. Finally, *C. leptoporus* was also observed in the Sub-Antarctic Zone of both transects, and additionally in the Antarctic Zone on Transect 1 (0.5-1.8 cells ml$^{-1}$).

Due the high relative contribution of *E. huxleyi* to the total coccolithophore community (>80% and often up to a 100%), surface trends in the distribution and abundance of total coccolithophores was

very similar to that of *E. huxleyi* (Figs. 4 and 5). Coccolithophore cells were completely absent from several stations to the south of the Southern Boundary (40, 43, 48 and 54), although low numbers of detached coccoliths (<0.01 × 10$^3$ coccoliths$^{-1}$ ml$^{-1}$) were present. Almost all detached coccoliths (~99%) came from *E. huxleyi* rather than from the other species present, with the average detached coccolith:cell ratio ~24:1. Coccosphere and detached coccolith distributions were very similar to each

other (Fig. 5A; Spearman's ρ = 0.97, p<0.01). Detached coccolith concentrations were maximal (12 × 10$^3$ mL$^{-1}$) between the SAF and southern branch of the PF (PF-S) on Transect 1, and the coccolith:cell ratio in this area was as high as 44:1. The maximum detached coccolith concentration on Transect 2 was associated with the SAF (8.5 × 10$^3$ mL$^{-1}$), and the coccolith:cell ratio here was 33:1. At the majority of sampling stations coccolithophore abundance was maximal in surface waters

(Charalampopoulou, 2011). However, at stations 18, 36, 62, 72 and 82 the maximum coccolithophore abundance was observed at 50 m depth (grey squares in Fig. 5A), although differences between surface and 50 m abundances were less than 60 cells mL$^{-1}$.

### 3.3     Community calcite production and cell-specific calcification

Bulk calcite production (CP) by the coccolithophore community in surface waters ranged between 0.3

and 18.6 μmol C m$^{-3}$ d$^{-1}$ (Fig. 5B). Relatively high CP was measured on either side of the SAF on Transect 1 (14.8-18.6 μmol C m$^{-3}$ d$^{-1}$). Unfortunately, CP measurements were not made at the station of maximum coccolithophore abundance (Stn 16). On Transect 2, maximum CP was measured at the SAF and just south of the Falkland Islands (~10 μmol C m$^{-3}$ d$^{-1}$). Minimum CP was measured close to the Antarctic Peninsula in the Continental Zone (<1 μmol C m$^{-3}$ d$^{-1}$). As with coccolithophore

abundance, CP was generally maximal at the surface (data not shown; Charalampopoulou, 2011) apart



from at stations 18, 36, 62, 72 and 82 where the maximum was observed at 50 m (grey squares in Fig. 5B). The difference between surface and 50 m CP was less than 1 μmol C m$^{-3}$ d$^{-1}$ at stations 18, 36, 62, and 82, whereas the difference was ~9 μmol C m$^{-3}$ d$^{-1}$ at station 72.

Cell-specific calcification (cell-CF) at the surface ranged from 0.01 to 0.16 pmol C cell$^{-1}$ d$^{-1}$ (Fig. 5C).

The highest values of cell-CF (0.13-0.16 pmol C cell$^{-1}$ d$^{-1}$) were observed in the Sub-Antarctic Zone on Transect 1, although lower values (0.04-0.06 pmol C cell$^{-1}$ d$^{-1}$) were also observed in this region. Cell-CF was generally less than 0.03 pmol C cell$^{-1}$ d$^{-1}$ south of the SAF on Transect 1. The exception to this was at station 36, just north of the SACCF, where cell-CF was ~2 pmol C cell$^{-1}$ d$^{-1}$ (data not shown in Fig. 5C) and a very small community (0.4 cells ml$^{-1}$) was evenly split between *E. huxleyi* and

*Pappomonas* spp. The CP at station 36 was also very low (0.8 μmol C m$^{-3}$ d$^{-1}$), and hence the cell-CF at this station should be viewed with caution (i.e., we ignore this station from our subsequent analysis).

On Transect 2, maximum cell-CF (0.10 pmol C cell$^{-1}$ d$^{-1}$) was observed just north of the SACCF. North of station 62, average cell-CF was ~0.05 pmol C cell$^{-1}$ d$^{-1}$ whereas values further south were lower. It was not possible to calculate cell-CF at station 54 because there was an absence of detectable

coccolithophore cells, despite a measurable, but very low, CP rate (0.4 μmol C m$^{-3}$ d$^{-1}$). Differences between surface and 50 m waters in terms of coccolithophore abundance and CP resulted in different cell-CF at 50 m (grey open squares in Fig. 5C) at stations 18, 36, 62, 72 and 82. However, none of these differences were significant relative to surface cell-CF (one-way ANOVA, $p < 0.001$).

### 3.4    Coccolith size, calcite content and production rates

The overall mean coccolith distal shield length (DSL) was 2.8 μm, while the median for each station ranged from 2.5 to 3.3 μm and the full range was 1.8 to 5.5 μm (Fig. 6A). Median DSLs were lowest in the Polar Frontal Zone on Transect 1 and in the Antarctic Zone on Transect 2. Maximum median DSLs were measured at stations located south of the Chile shelf (Stations 3, 5, 8 and 11), and were significantly larger (pair-wise Turkey tests, $p < 0.05$) than at all the other stations. Minimum median

DSL was measured at station 62 (2.5 μm). When DSLs were converted to coccolith calcite content, following the equations of Young and Ziveri (2000), the overall mean was 0.010 pmol coccolith$^{-1}$ and the median for each station ranged between 0.007 and 0.015 pmol coccolith$^{-1}$ (Fig. 6B), with the full range from 0.003 to 0.035 pmol coccolith$^{-1}$. Due to the method of estimating coccolith calcite content, and the complete dominance of morphotype B/C across Drake Passage, it showed an almost identical

latitudinal pattern as DSL (Figs. 6A and 6B).

Division of cell-CF by estimates of coccolith calcite content gives an estimate of the rate of coccolith production per cell (Poulton et al., 2010, 2013). For communities across Drake Passage the overall mean coccolith production rate was 6 coccoliths cell$^{-1}$ d$^{-1}$ (Fig. 6C). Coccolith production rates were not calculated for stations 36 and 54 (see section above), despite the presence of detached coccoliths




(Figs. 5A, 7A and 7B). Coccolith production rates were significantly lower south of 59ºS (<2 coccoliths cell$^{-1}$ d$^{-1}$ in Continental and Antarctic Zones, Stns 22 - 58) than in the Polar Frontal Zone, the Sub-Antarctic Zone and at stations 62 and 65 in the northern part of the Antarctic Zone on Transect 2 (3-18 coccoliths cell$^{-1}$ d$^{-1}$) (Kruskal-Wallis, p<0.001; pairwise Tukey tests , p<0.05). The highest coccolith production rates were observed at stations 14 and 62 (16-18 coccoliths cell$^{-1}$ d$^{-1}$).

**3.5    Relationships to environmental variables**

A multivariate analysis of variability in the coccolithophore community compared to that of environmental conditions (a BEST test) showed the strongest Spearman's rank correlation with a combination of temperature and $\bar{E}_{MLD}$ (ρ = 0.393, p<0.01). Spearman's rank correlations were also carried out between the Principal Components (PC-1 and PC-2), individual environmental variables,

coccolithophore diversity, CP, cell-CF, coccolith calcite content and coccolith production rates (Table 2). PC-1 was significantly correlated with coccolith calcite content, cell-CF and CP (p<0.01) as well as species number and coccolith production rates (p<0.05). In contrast, correlations between PC-2 and characteristics of the coccolithophore community and cellular calcification were not statistically significant. Species richness (i.e. the number of species present) showed significant (p<0.05) positive

correlations with temperature, salinity, $\Omega_C$, $\bar{E}_{MLD}$ and PAR$_{above surface}$ , and negative correlations with PC-1 and nutrient concentrations. Coccolith calcite content and CP were positivity correlated (p<0.05) with temperature and $\Omega_C$ and negatively correlated (p<0.05) with PC-1 and nutrient concentrations. Cell-CF and coccolith production rates also showed a similar trend of statistically significant (p<0.05) correlations, although they were additionally correlated with $\bar{E}_{MLD}$ and PAR$_{above surface}$ (Table 2).

Generally, strong north-south gradients in environmental variables (temperature and $\Omega_C$ decrease towards the south while nutrient concentrations increase), were also evident in coccolithophore diversity and cellular calcification across Drake Passage.

**4    Discussion**

**4.1    Coccolithophore distribution in the Southern Ocean**

In this study, the surface waters across Drake Passage were sampled along two latitudinal transects to assess coccolithophore abundance and species distribution. The total coccolithophore abundances that we observed (up to ~600 cells mL$^{-1}$) agree with previous observations of maximum abundances of between 200 and 500 cells mL$^{-1}$ in the Atlantic, Pacific, Indian and Australian sectors of the Southern Ocean (Cubillos et al., 2007; Eynaud et al., 1999; Findlay and Giraudeau, 2000; Gravalosa et al.,

2008; Mohan et al., 2008; Hinz et al., 2012). A previous study across Drake Passage, coinciding with the eastern transect (Transect 2), reported similar abundances of up to 600 cells mL$^{-1}$ (Holligan et al., 2010). Maxima in coccolithophore abundance were associated with oceanic fronts and particularly with the SAF and PF, as observed in other studies (Eynaud et al., 1999; Gravalosa et al., 2008;



Holligan et al., 2010). These abundance maxima may be related to high primary productivity due to

the dynamics of frontal systems and increased nutrient (presumably dissolved iron) availability
(Eynaud et al., 1999; Gravalosa et al., 2008).

The southern limit for coccolithophores was once thought to be the PF (Winter et al., 1999); however, low numbers of coccolithophores have more recently been observed as far south as the SACCF (Eynaud et al., 1999; Findlay and Giraudeau, 2000; Cubillos et al., 2007; Gravalosa et al., 2008;

Mohan et al., 2008; Hinz et al., 2012; Winter et al., 2014). In the Southern Ocean, temperature has been suggested to control coccolithophore distribution, as coccolithophore barren waters are typically less than 2ºC (Holligan et al., 2010). In our study, no *E. huxleyi* cells were found south of the SACCF in surface waters, although detached coccoliths were still present (<20 coccoliths mL$^{-1}$) and low numbers of *E. huxleyi* (<13 cells mL$^{-1}$) were observed in deeper waters (>25 m). Low abundances (1-2

cells mL$^{-1}$) of *W. antarctica* were also found in samples south of the SACCF and at the Southern Boundary of the ACC. However, during our study surface water temperatures were always above 2ºC, even at the most southerly stations.

A number of previous studies have reported a succession of *E. huxleyi* morphotypes with increasing latitude in the Southern Ocean, with morphotype A being replaced by morphotype B/C towards

Antarctica in the Australian and Indian sectors (Findlay and Giraudeau, 2000; Cubillos et al., 2007; Mohan et al., 2008; Cook et al., 2011). Across Drake Passage we only observed morphotype B/C, regardless of latitude. Sea surface temperatures were always less than 10ºC across Drake Passage, conditions which seem to favour dominance of the B/C morphotype (Findlay and Giraudeau, 2000; Mohan et al., 2008; Holligan et al., 2010; Poulton et al., 2011; Cook et al., 2011). In the Australian

sector of the Southern Ocean, morphotype A occurs as far south as the Sub-Antarctic Zone (Cubillos et al., 2007), while in the Atlantic sector it has only been observed in warm, nutrient poor waters north of the Patagonian Shelf so far (Poulton et al., 2011).

Apart from *E. huxleyi*, other coccolithophore species are scarce in the Southern Ocean, especially pole-wards of the Sub-Antarctic Zone (Eynaud et al., 1999; Mohan et al., 2008; Holligan et al., 2010).

In this study, *G. muellerae* was found at moderate abundances (up to 36% of total numbers) south of the Falklands and close to the SAF. These observations match those of Holligan et al. (2010) in the same area and suggest that *G. muellerae* is a characteristic species in the region south of the Falkland Islands. *Gephyrocapsa muellerae*, *A. quattrospina*, *C. caudatus* and *C. leptoporus* were also more abundant north of the Polar Front (Fig. 4), which confirms their preference for sub-polar regions in

both the Southern Ocean (Eynaud et al., 1999; Findlay and Giraudeau, 2000) and the Northern Hemisphere (Samtleben et al., 1995; Baumann et al., 2000). *Pappomonas* spp. and *W. antarctica*, with their small cells and low calcite-containing coccoliths (Young et al., 2003), were ubiquitous and the only species recorded in surface waters of the Continental Zone close to Antarctica, albeit at low cell



densities. Species of the family Papposphaeraceae (including *Pappomonas* and *Papposphaera* spp.)

are characteristic of Arctic waters (Thomsen, 1981; Charalampopoulou et al., 2011) and, together with

*Wigwamma* spp., they are also characteristic of the Australian Antarctic Zone (Findlay and Giraudeau,

2000).

### 4.2   A pole-wards decrease in calcification?

A conspicuous feature of the results as a whole is the southwards decline in coccolithophores

(abundance, diversity) and calcite production (Fig. 5). Most parameters have higher average values

closer to South America and lower average values closer towards Antarctica. The prevalence of this

trend was examined further by splitting the data from the intensively-studied stations (red circles in

Fig. 1) into two groups, a more northerly one consisting of data north of the polar front (north of PF(S)

on the western transect) and a more southerly one consisting of data south of the polar front (stations

36 to 55). One-tailed, two-sample *t*-tests (unpaired, unequal sample sizes and unequal variances)

support the hypothesis that values were, on average, significantly higher to the north (i.e. rejected the

null hypothesis that values to the south were higher than or equal to values to the north) for most of the

parameters shown in Figure 5 (coccosphere concentration, p<0.001; coccolith concentration, p <0.01;

calcite production rate, p<0.001; cellular calcification rate, H0 not rejected). Coccolithophore and

coccolith concentrations, as well as community calcite production, were all found to be lower on

average to the south. However, the hypothesis of significantly lower values to the south was not

supported for the rates of cell-specific calcification (cell-CF) or coccolith production. Some values of

these parameters are high in the Antarctic Zone on the eastern transect (Figs. 5C and 6C), where non-

negligible calcite production (Fig. 5B) were measured at stations 62 and 65 in waters containing very

few cells (Fig. 5A). These surprising higher values could potentially be explained by either (i)

coccolithophores calcifying at relatively normal rates to the south (see below); or (ii) artefacts in the

calculations due to uncertainties from low cell number counts. However, calculations of propagated

errors (shown in Fig. 5C) suggest that such errors only explain a small proportion of the rates of cell-

CF and coccolith production at stations 62 and 65.

The absence from high-latitude polar waters was, until recently, a long-held paradigm in

coccolithophore ecology (McIntyre & Bé, 1967; Winter & Siesser, 1994), although more recent

studies have found a pole-wards expansion in their range (e.g., Holligan et al., 2010; Winer et al.,

2014). When combined with the pole-wards decrease in the $CaCO_3$ saturation state of seawater (Orr et

al., 2005), this led to the suggestion (e.g., Cubillos et al., 2007; Beaufort et al., 2011) that

coccolithophores may struggle in polar waters because of difficulties in calcifying in the low $CaCO_3$

saturation conditions. The data presented here are, however, not consistent with this hypothesis. If low

saturation states inhibit calcification, and this is a dominant reason for low coccolithophore abundance

where it occurs, then we would expect to see low abundances in the same places as we see low cell-



CF. This is true in some areas; for instance, both parameters have very low values towards the
southern ends of both transects (e.g. stations 41 and 58; Fig. 5). However, there are also areas where
coccolithophores are scarce but cell-CF are high (for instance stations 62 and 65; Fig. 5) and,
conversely, where coccolithophores are relatively numerous but cell-CF is low (for instance stations
22 and 27; Fig.5). Overall these two parameters were in fact anti-correlated in surface waters of the
study area (Spearman ρ = -0.46, p<0.05), suggesting that difficulty in calcifying may not be the main
reason for the very low coccolithophore numbers in some parts of the Southern Ocean. Rather,
variability in physiological factors (e.g., light availability), which influence intrinsic growth rates, and
mortality factors (e.g., grazing rates) may be more influential on coccolithophore cell numbers
(Poulton et al., 2010). Species-specific differences in cellular calcite content and variability in intrinsic
growth rates will also influence trends in cell-CF (Poulton et al., 2010; Charalampopoulou et al.,
515    2011).

### 4.3    Cellular calcification across Drake Passage

This study presents the most southerly direct measurements of calcification rates currently available.
*Emiliania huxleyi* morphotype B/C represented over 80% of total cell numbers (and up to a 100% in
many cases) across Drake Passage, with other species (*A. quattrospina*, *C. leptoporus*, *G. muellerae*)
contributing relatively little to the total community. Hence, we propose that the calcification rates
presented in this study are characteristic of *E. huxleyi* across Drake Passage. Calcite production (CP)
was low (<20 $\mu$mol C m$^{-3}$ d$^{-1}$) compared to subarctic regions during both bloom (50-1500 $\mu$mol C m$^{-3}$
d$^{-1}$; Fernandez et al., 1993; Poulton et al., 2013, 2014) and non-bloom conditions (50-250 $\mu$mol C m$^{-3}$
d$^{-1}$; Poulton et al., 2010), but more similar to measurements from the (sub-)tropics (10-50 $\mu$mol C m$^{-3}$
d$^{-1}$; Poulton et al., 2007; Balch et al., 2011). The ratio of CP to primary production for the bulk
phytoplankton community (data not shown) was also similar to (sub-) tropical communities (0.001-
0.069; Poulton et al., 2007), highlighting that coccolithophores contributed only a small fraction
(<1%) to upper ocean carbon fixation across Drake Passage (see Charalampopoulou, 2011).

The range of cell-CF was also lower (0.01-0.16 pmol C cell$^{-1}$ d$^{-1}$) than that found in the Iceland Basin
during non-bloom conditions (0.25-0.75 pmol C cell$^{-1}$ d$^{-1}$; Poulton et al., 2010), or in mono-specific *E.
huxleyi* cultures (0.2-0.8 pmol C cell$^{-1}$ d$^{-1}$; Balch et al., 1996). However, cell-CF was more similar to
measurements from the Patagonian Shelf bloom (0.05-0.6 pmol C cell$^{-1}$ d$^{-1}$; Poulton et al., 2013). In
the Iceland Basin, *E. huxleyi* morphotype A dominates, whereas on the Patagonian Shelf and across
Drake Passage morphotype B/C dominates. These levels of cell-CF translate into coccolith production
rates of 0.1-1.2 coccoliths cell$^{-1}$ h$^{-1}$ (2-18 coccoliths cell$^{-1}$ d$^{-1}$, assuming a 15-h light period), using the
mean coccolith calcite values estimated for morphotype B/C at each station (0.007-0.015 pmol C
coccolith$^{-1}$). These are within the range reported for morphotype A in culture (0-3 coccoliths cell$^{-1}$ h$^{-1}$;
Balch et al., 1996) and field studies in the Iceland Basin (0.4-1.8 coccoliths cell$^{-1}$ h$^{-1}$; Fernandez et al.,



1993; Poulton et al., 2010). Measurements from the Patagonian Shelf bloom, composed exclusively of

*E. huxleyi* morphotype B/C, gave coccolith production rates of 0.1-3.3 coccoliths cell$^{-1}$ h$^{-1}$ (Poulton et al., 2013). Hence, even though cell-CF rates for morphotype B/C appear to be low compared to morphotype A, the rate at which individual coccoliths were produced is generally similar between *E. huxleyi* morphotypes when differences in coccolith calcite content are taken into account.

Although there is general similarity in the range of coccolith production rates between sub-polar

environments, the calculated rates along the two transects (Fig. 6C) show a general trend across Drake Passage of lower coccolith production close to Antarctica in the Continental Zone than further north (Fig. 6C) (although stations 62 and 65 are, as noted above, an exception to the north-south trend). Hence, indices of cellular calcification (including cell-specific calcification, calcite content per coccolith, and coccolith production rates), all show reductions south of the Sub-Antarctic Front in

Drake Passage and with increasing proximity to the Antarctic Peninsula. It is important to clarify that the decrease in cell-CF is linked to reductions in the rate at which each cell produces coccoliths rather than strong reductions in the size or calcite content of individual coccoliths: coccolith calcite is a poor indicator of coccolith production rates or levels of cellular calcification. Apart from those close to Antarctica, *E. huxleyi* populations in the Southern Ocean are producing coccoliths at similar rates to

populations in other environments.

### 4.4 Coccolith morphometrics across Drake Passage

In this study, the overall mean DSL of *E. huxleyi* morphotype B/C detached coccoliths (2.8 µm; see Fig. 6) was at the lower end of the range reported in cultured B/C strains (range 2.65-4.80 µm; Cook et al., 2011), and lower than the average seen on the Patagonian Shelf (3.25 ± 0.40 µm; Poulton et al.,

2011). A strong latitudinal trend was observed in DSL of morphotype B/C coccoliths, from the Chile shelf (median >3.1 µm) to smaller ones further south (median <2.9 µm). A north-south trend was also observed from 'over-calcified to weakly calcified *E. huxleyi* morphotypes' in the Australian and Indian sectors of the Southern Ocean (Findlay and Giraudeau, 2000; Cubillos et al., 2007; Mohan et al., 2008). However, a similar trend was not observed south of the Falklands (station 82; Fig. 6), where

average DSL was also <2.9 µm, even though environmental conditions were similar to those off Chile.

Coccolith calcite content derived from our DSL measurements (mean 0.010 pmol C coccolith$^{-1}$) was lower than estimates for the B/C morphotype based on a regression of total coccoliths to discrete calcite measurements (0.020 pmol C coccolith$^{-1}$; Holligan et al., 2010), and from DSL measurements of detached coccoliths on the Patagonian Shelf (0.015 pmol C coccolith$^{-1}$; Poulton et al., 2011). In our

study, coccolith calcite was between 0.013 and 0.015 pmol C coccolith$^{-1}$ off southern Chile and <0.009 pmol C coccolith$^{-1}$ across the rest of Drake Passage. The wide range of DSL observed across Drake



Passage (1.8-4.4 μm), also highlights the natural variability found within populations of *E. huxleyi* (Poulton et al., 2011; Young et al., 2014).

### 4.5    Environmental drivers of coccolithophores and calcification

The combination of environmental variables best able to explain variability in coccolithophore community composition across Drake Passage were temperature and mixed layer irradiance ($\rho$ = 0.393, p<0.05). These were also the two variables best correlated with the first and second principal components (PC-1 and PC-2) in the PCA of environmental factors. Combined PC-1 and PC-2 explained 80% of the variation in the environmental data. Mohan et al. (2008) also found that

coccolithophore distribution south of Madagascar towards Antarctica was controlled by temperature and light, with higher diversity in warmer and higher irradiance conditions, and high abundances of mono-specific *E. huxleyi* assemblages corresponding to high nitrate concentrations in the Sub-Antarctic Zone. Across Drake Passage, macronutrient concentrations were not limiting, while $\bar{E}_{MLD}$ was less than 3 mol PAR m$^{-2}$ d$^{-1}$ across the Continental Zone and just north of the SAF on Transect 2,

a threshold below which light is potentially limiting for Southern Ocean phytoplankton (Venables and Moore, 2010). This suggests that coccolithophore distribution and abundance is controlled primarily by temperature and light rather than nutrient concentrations or carbon chemistry.

Calcification parameters (coccolith calcite content, coccolith production rates, cell-CF and CP) were all negatively correlated with PC-1, and were higher in warm, lower nutrient, higher $\Omega_C$ conditions;

they all decreased towards Antarctica, as the individual correlations show (Table 2). Additionally, coccolith production rates and cell-CF were positively correlated with $\bar{E}_{MLD}$ and both of these were higher at stations where both temperature and $\bar{E}_{MLD}$ were relatively high (mean values of 6.6°C and 7.4 mol PAR m$^{-2}$ d$^{-1}$, respectively). The positive correlation of coccolith production rates and cell-CF with $\bar{E}_{MLD}$ is not surprising as these are strongly light dependent in *E. huxleyi* cultures (Linschooten et al.,

1991; Zondervan et al., 2002), and in field populations where both total and cell-CF rates decrease with depth (Fernandez et al., 1993; Poulton et al., 2010, 2014).

The correlation of calcification indices with phosphate and nitrate is most likely the result of the strong anti-correlations that exist between nutrient concentrations and temperature ($r^2$ = 0.88 and 0.89, nitrate and phosphate respectively; n = 50, p<0.01) across Drake Passage. A similar inter-correlation may

also explain the observed trend of decreasing calcite production with decreasing $\Omega_C$ across Drake Passage, as this variable is also strongly correlated with temperature ($r^2$ = 0.86, n = 50, p<0.01). A strong trend in cellular calcification (cell-CF, coccolith production) with $\Omega_C$ is in general agreement with suggestions of carbonate chemistry as a strong driver of pelagic calcification (Cubillos et al., 2007; Beaufort et al., 2011). However, as can be seen across Drake Passage, $\Omega_C$ co-varies with all the

other growth-dependent variables (i.e., nutrients, irradiance, and temperature) and the effect of one



over any of the others is difficult to distinguish. This highlights how environmental gradients in the ocean can work in tandem to control coccolithophore growth and calcification: organisms are not usually faced with only one eco-physiological driver changing at a time, the environment is multivariate.

The one growth-dependent factor lacking from our study, especially relevant to the Southern Ocean, is iron availability. Notably, within the factors included only a limited degree of the variability in coccolithophore species distribution ($\rho = 0.393$) was explained and only moderate correlations between calcification and environmental factors were found (Table 2). Presently, the influence of iron on coccolithophore distribution and physiology is unclear and the literature is conflicting. Brand

(1991) found *E. huxleyi* to have relatively low iron requirements compared with diatoms (Zondervan et al., 2007), with some iron addition experiments showing no response (Lam et al., 2001; Assmy et al., 2007). In contrast, other studies have shown increased CP (Crawford et al., 2003) and cell abundances (Nielsdóttir et al., 2009) and proposed iron as a potentially important growth-limiting factor to sub-polar coccolithophore communities (Poulton et al., 2010). Furthermore, iron supplied

through sedimentary sources has also been suggested to influence the formation of the Patagonian Shelf coccolithophore bloom, which occurs in cold, macronutrient rich water originating north of the SAF (Garcia et al., 2008; Poulton et al., 2013, Balch et al., 2014). Clearly more work is required to examine the role of iron on Southern Ocean coccolithophore biogeography and calcification, as well as its potential influence on coccolithophore blooms and coccolithophore growth in mixed

communities.

### 4.6 Global change effects on coccolithophores in the Southern Ocean

Future changes in the Southern Ocean are expected to include higher temperatures and stronger stratification, leading to higher $\bar{E}_{MLD}$ but lower nutrient availability (Boyd et al., 2008), while at the same time $\Omega_C$ is predicted to decrease (Hauri et al., 2015). Our statistical analysis suggests that

changes to temperature and/or $\Omega_C$ both affect coccolithophores, but we were not able to determine robustly which out of these is likely to be most critical. In addition, higher $\bar{E}_{MLD}$ is predicted to favour coccolithophores and more recently increased $pCO_2$ has been suggested to favour the pole-wards expansion of *E. huxleyi* (Winter et al., 2014) and increases in coccolithophores in the North Atlantic (Rivero-Calle et al., 2015). Across Drake Passage, increases in temperature and stratification could

therefore potentially facilitate a pole-wards expansion of the range of *E. huxleyi*, which would in turn reduce the extent of the low-calcite production area around Antarctica. Such a pole-ward migration of *E. huxleyi* might lead to a range expansion of the B/C morphotype, the low calcite ecotype found in the Southern Ocean (this study; Cook et al., 2011, 2013; Poulton et al., 2011, 2013).





Although coccolithophore cell densities may increase, with coccolith production rates fairly similar to

North Atlantic populations, if B/C remains the dominant morphotype then the resulting calcification

rates will still be relatively low due to the low cellular calcite. Such a migration of *E. huxleyi* has

already been observed in the Australian sector of the Southern Ocean (Cubillos et al., 2007; Winter et

al., 2014), as well as in the Bering and Barents seas (Merico et al., 2003; Smyth et al., 2004). Limited

sensitivity to $CO_2$ has also been observed in several strains of *E. huxleyi* (Langer et al., 2009, 2011)

and at low growth irradiances (<5 mol PAR $m^{-2}$ $d^{-1}$; Zondervan et al., 2002). In the Southern Ocean,

$\bar{E}_{MLD}$ is generally low (mean ~ 5 mol PAR $m^{-2}$ $d^{-1}$ in this study) and short-term predicted changes are

unlikely to exceed background variability (Boyd et al., 2008). Hence, it is also feasible that any

reductions in calcification due to ocean acidification may be minimal or be opposed by the effects of

increased sea-surface temperatures and light availability. Furthermore, if increased $CO_2$ availability for

*E. huxleyi* also favours growth and range expansions of this species (Winter et al., 2014; Rivero-Calle

et al., 2015), then this will also compound the potential effects of changes in the environment.

### 4.7    Summary

The coccolithophore community across Drake Passage was dominated by the low coccolith-calcite

B/C morphotype of *E. huxleyi*. Most coccolithophore and calcification indices declined towards

Antarctica, including both bulk and individual rates of calcification per cell. Despite this,

coccolithophore abundance and individual calcification rate per cell were anti-correlated. Taken as a

whole, measures of coccolithophore abundance and calcification across Drake Passage were rather

low compared to elsewhere in the global ocean, although cellular rates of coccolith production (0.1-1.2

coccoliths $cell^{-1}$ $h^{-1}$) were very similar to values in the Iceland Basin (Poulton et al., 2010) and on the

Patagonian Shelf (Poulton et al., 2013), except at the southern end of the transects where they declined

to very low values (<0.3 coccoliths $cell^{-1}$ $h^{-1}$). However, due to the low coccolith calcite content

characteristic of morphotype B/C, community calcite production (CP) (<20 $\mu$mol C $m^{-3}$ $d^{-1}$) was more

similar to rates in (sub-) tropical waters than in the Iceland Basin (Poulton et al., 2007, 2010; Balch et

al., 2011).

Temperature and irradiance were found to be best able to explain variation in coccolithophore species

distribution and abundance across Drake Passage. Similarly, calcification parameters correlated with

the strong latitudinal gradients in temperature, nutrients and $\Omega_C$, while cell-specific calcification and

coccolith production rates also correlated with $\bar{E}_{MLD}$. However, temperature, nutrients and $\Omega_C$ were all

strongly inter-related across Drake Passage, and so it was not possible to robustly separate their

individual influence on the calcification parameters. It is therefore difficult to be sure how

coccolithophores and calcite production in the Southern Ocean will respond to global change as a

whole, because of the contrasting predicted trends in water temperature, nutrients and $\Omega_C$.



**Acknowledgements.** We would like to thank the officers and crew of the RRS *James Clark Ross*, the scientists and Principal Scientific Officer, Elaine McDonagh, of JC031. Furthermore, we would like to

thank Chris Daniels for reading and commenting on a previous draft. The authors would also like to acknowledge the support of Oceans 2025 and National Capability funding which supports the Drake Passage Sustained Observation project. AC was supported by a University of Southampton PhD studentship, with her participation in the JC031 supported by Oceans 2025 funding. AJP was supported by a UK Natural Environmental Research Council (NERC) postdoctoral fellowship

(NE/F015054/1). DCEB was supported by a NERC research grant (NE/F01242X/1). AC and TT also acknowledge financial support from the 'European Project on Ocean Acidification' (EPOCA) which received funding from the European Community's Seventh Framework Programme (FP7/2007-2013) under grant agreement no. 211384.

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



Table 1. Principal Component (PC) analysis and Pearson product moment correlations between PC
scores and environmental variables. ns indicates not significant.


|  | PC-1 (62.4%) | PC-2 (18.2%) |
|---|---|---|
| *Eigenvectors* | | |
| Temperature | -0.44 | -0.03 |
| Salinity | -0.33 | 0.29 |
| Nitrate | 0.43 | -0.07 |
| Phosphate | 0.43 | -0.04 |
| pH | -0.07 | 0.58 |
| $\Omega_C$ | -0.43 | 0.20 |
| $\bar{E}_{MLD}$ | -0.21 | -0.60 |
| PAR $_{above\ surface}$ | -0.32 | -0.40 |
| *Environmental variables* ($p<0.01$) | | |
| Temperature | -0.97 | ns |
| Salinity | -0.75 | ns |
| Nitrate | 0.95 | ns |
| Phosphate | 0.95 | ns |
| pH | ns | 0.70 |
| $\Omega_C$ | -0.95 | ns |
| $\bar{E}_{MLD}$ | -0.46 | -0.73 |
| PAR $_{above\ surface}$ | -0.72 | -0.49 |



Table 2. Spearman's rank correlations of species richness, calcification parameters and environmental

variables. PC-1 is a combination of temperature, phosphate, nitrate and $\Omega_C$ and PC-2 is a combination

of $\bar{E}_{MLD}$ and pH. na indicates not applicable, ns not significant.

| Environmental variables | Species Richness | CP | Cell-CF | Coccolith calcite content | Coccolith production rate |
|---|---|---|---|---|---|
| PC-1 | -0.70[a] | -0.64[b] | -0.74[b] | -0.57[b] | -0.66[a] |
| PC-2 | ns | ns | ns | ns | ns |
| Temperature | 0.75[a] | 0.73[b] | 0.64[b] | 0.61[b] | 0.58[b] |
| Salinity | 0.64[a] | 0.57[a] | ns | ns | ns |
| Nitrate | -0.63[a] | -0.57[a] | -0.75[b] | -0.63[b] | -0.67[b] |
| Phosphate | -0.64[a] | -0.56[a] | -0.80[b] | -0.60[b] | -0.72[b] |
| pH | Ns | ns | ns | ns | ns |
| $\Omega_C$ | 0.70[a] | 0.67[b] | 0.70[b] | 0.57[a] | 0.63[b] |
| $\bar{E}_{MLD}$ | 0.29[a] | ns | 0.62[b] | ns | 0.61[b] |
| PAR $_{above\ surface}$ | 0.40[a] | ns | 0.50[a] | ns | 0.49[a] |
| Temperature, $\bar{E}_{MLD}$ | na | na | na | na | na |

[a] $p<0.05$; [b] $p<0.01$.



FIGURE LEGENDS

Fig. 1. JC31 cruise track, showing Transect 1 (Chile to Antarctica) and Transect 2 (Antarctica to

Falklands). Blue and red circles indicate sampling stations. Red circles are numbered and indicate

stations where calcification rates were also measured. The locations of the following fronts are shown

on each transect: Subantarctic Front (SAF), Polar Front (PF), Southern Antarctic Circumpolar Current

Front (SACCF) and Southern Boundary of the ACC (SB). Where two possible locations or two

branches of a front were observed, these are denoted with a northern (N) or southern (S) suffix.

Fig. 2. Surface distribution of physicochemical environmental variables along Transect 1 (left) and

Transect 2 (right). (A) Temperature and salinity. (B) Nitrate, silicate and phosphate concentrations.

(C) $pH_T$ and calcite saturation state ($\Omega_C$). (D) Euphotic zone depth ($Z_{eu}$), mixed layer depth (MLD) and

Chl-*a*. (E) Above surface irradiance ($PAR_{above\ surface}$) and mixed layer irradiance ($E_{MLD}$).

Fig. 3. Ordination from Principal Component Analysis of environmental variables. Environmental

gradients are displayed as arrows indicating the direction of greatest change. Filled symbols represent

samples from the different zones of Transect 1, and empty symbols samples from Transect 2. SAZ =

Subantarctic zone, PFZ = Polar Frontal Zone, AZ = Antarctic Zone, CZ = Continental Zone. (N) and

(S) denote the northern and southern part of the AZ on Transect 1, as a result of the two branches of

the Polar Front. Arrows and length of the arrows indicate the relative influence of each environmental

variable on the PC axis; e.g., variables which align with PC1 (temperature, phosphate, nitrate) strongly

influence PC1. Arrows which go in opposite directions indicate opposing relationships; e.g.,

temperature and nutrient concentrations are negatively related.

Fig. 4. Abundance of major coccolithophore species along Transect 1 (left) and Transect 2 (right).

Note the different scales of the abundance axes for different species.

Fig. 5. Surface distribution of coccolithophore variables along Transect 1 (left) and Transect 2 (right).

(A) Total coccolithophore and detached coccolith abundance. (B) Community (bulk) calcification

rates. (C) Cell-specific calcification rates. Grey filled and open squares show coccolithophore



abundance, calcification and cell-specific calcification at 50 m depth, if the maximum was observed at

this depth.

Fig. 6. Box-whisker plots showing, for *Emiliania huxleyi* only, the size distribution of (A) Coccolith

distal shield length ($n = 50$), (B) Coccolith calcite content and (C) Surface coccolith production rates

per cell, in each of twenty stations. The solid line within boxes indicate the median, with 5$^{th}$ and 95$^{th}$

percentiles of the data bounding the box, horizontal bars indicate the minimum and maximum values

and the points represent the 5th and 95th percentiles. The overall average for the whole dataset is

indicated by the horizontal dashed lines across the panels. Asterisks indicate stations where maximum

coccolithophore abundance and CP was measured at 50 m rather than in the surface.



Charalampopoulou et al.

Figure 1.

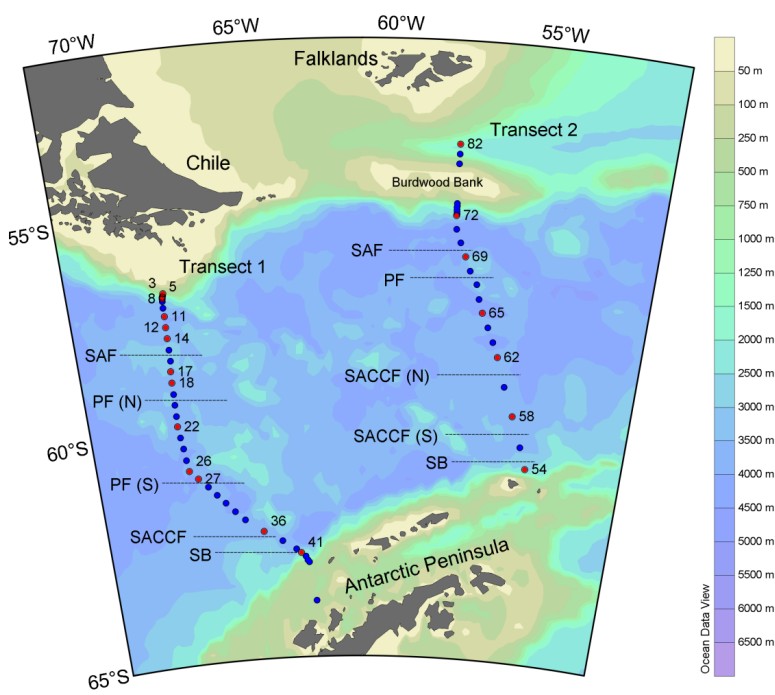





Charalampopoulou et al.

Figure 2.

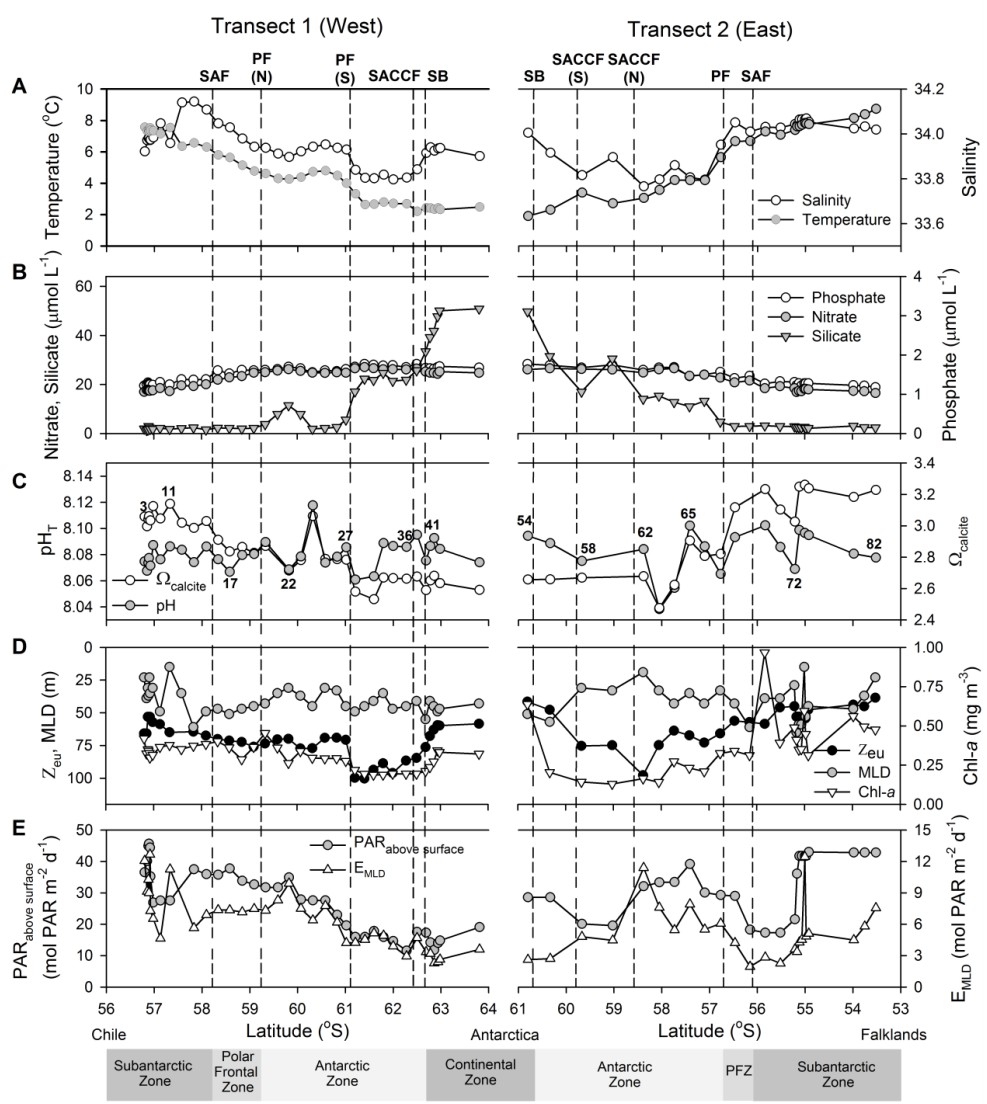





Charalampopoulou et al.

Figure 3.

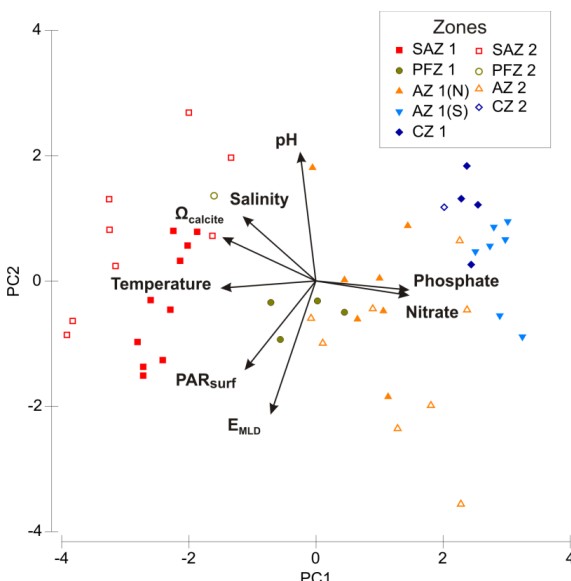





Charalampopoulou et al.

Figure 4.

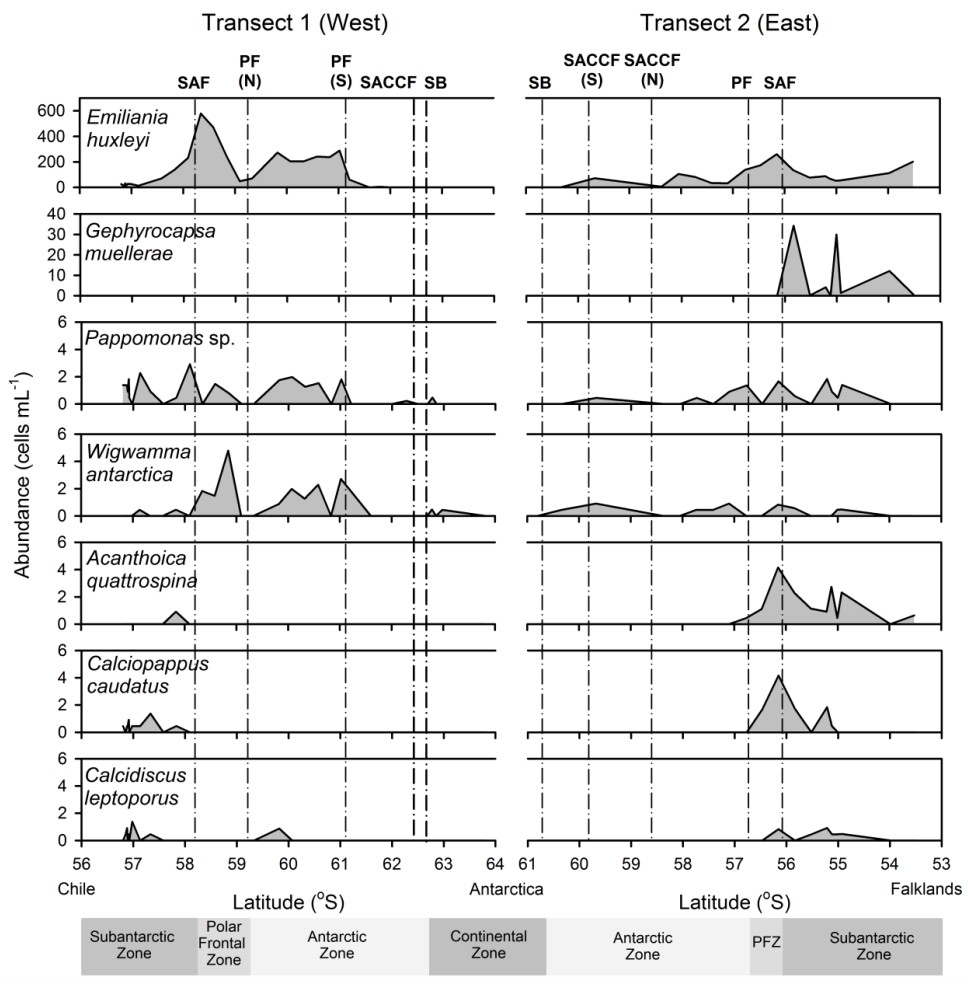





Charalampopoulou et al.

Figure 5.

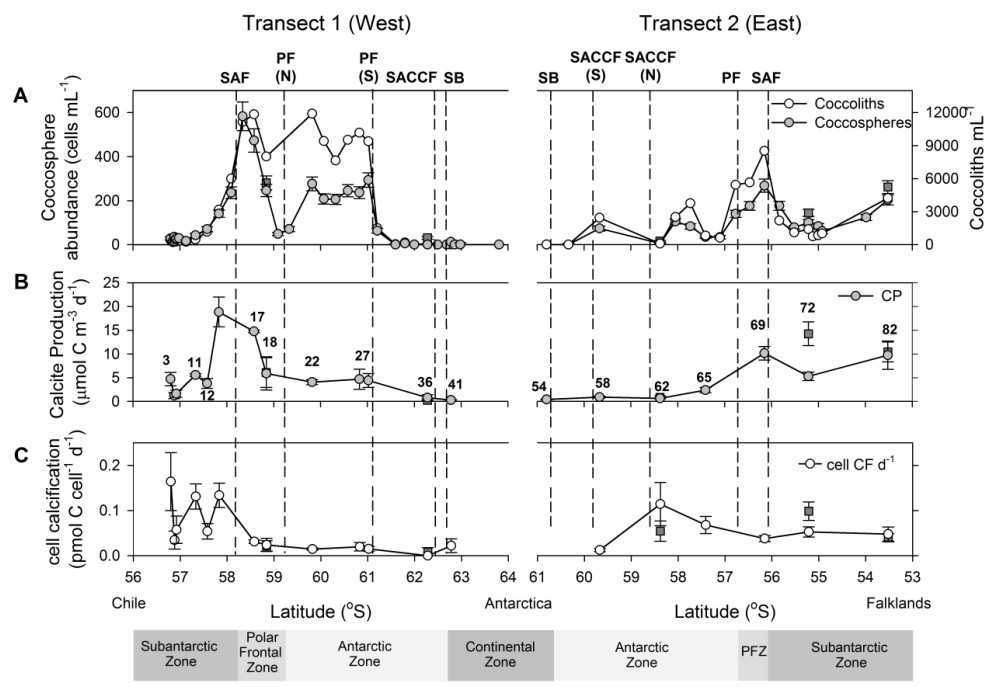





Charalampopoulou et al.

Figure 6.

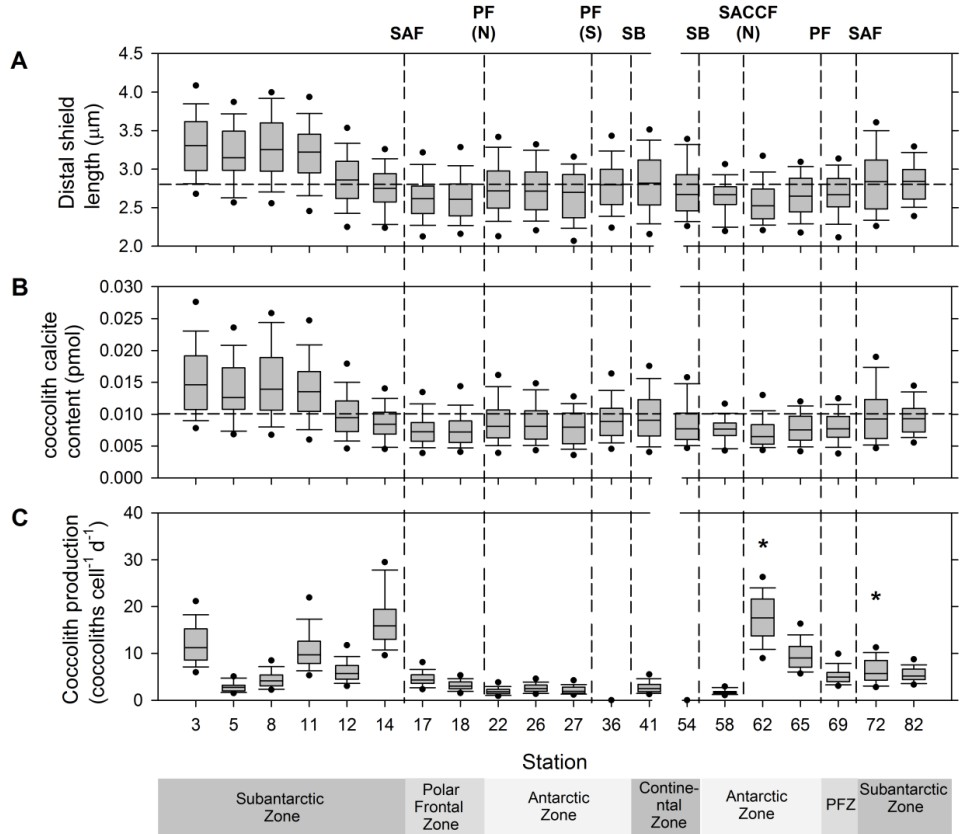