# Peer review of "Environmental drivers of coccolithophore abundance and calcification across Drake Passage (Southern Ocean)"

_Biogeosciences, 2016_

## Referee Comment (RC1) · Anonymous Referee #1 · 10 May 2016

This paper is well written and presents new data on coccolithophore distribution and calcite production in the Drake Passage. There is also a lot of accompanying physical and chemical data that are not usually taken on such cruises. Although conclusions are not definitive they will contribute to the ongoing discussion of the role of coccolithophores in the anthropocene.

Line 13 Although coccolithophore diversity or abundance?

Line 19 Although CP represents less than 1% of total carbon fixations How does this compare to what has been written about the Great Calcite Belt? How does the unique doubling time of coccolithophores influence the equation?

[Figure]

Line 71 Length of (what) exposure.

Line 113 Sampling What is the importance of the temporal timing of the sampling? Would it make a difference? Jan or March?

Line 117 samples were collected from the upper 100m of the water column Is it possible to get the depths of the samples? Later line 139 "5 CTD depths over the upper 100 m". Also in Figure 4: line 949 Abundance of Major coccolithophore species Which depths?...Not clearly stated the difference between Fig 5 surface distribution of coccolithophores and Fig 4.

Line 136 as well as the criteria of Orsi et al What criteria?

Line 320 How do the coccolithophores assemblages found in the Drake Passage compare to other publications of coccolithophores in the southern ocean?

Line 322 Drake Passage: including???? "Including" is not a clear word.

Line 430 A previous study across the Drake Passage When was this study undertaken?

Line 497 Winer et al Winter et al

Line 528 coccolithophoroes contributed only a small fraction What were the main contributors

---

## Referee Comment (RC2) · M. Saavedra-Pellitero (Referee) · 18 May 2016

The present study by Charalampopoulou et al. addresses the composition and structure of coccolithophore communities and calcite production across Drake Passage (Southern Ocean), regarding also primary production, chlorophyll-a, nutrient concentration, temperature, salinity, irradiance and carbonate chemistry parameters. The manuscript is well written and adds an interesting contribution to the ecology of coccolithophores at high latitudes in a marked environmental N-S gradient. I consider that this manuscript is novel and addresses compelling scientific questions within the scope of Biogeosciences.

Specific comments:

[Figure]

L. 44: What about morphotype C described in Young et al. (2003)?

L. 97: Would it be possible to add other references here on top of Winter et al. (2014)?

L. 123: Section 2.2 Study area should go before section 2.1 Sampling.

L. 117. Is it possible to know at which depths (0-100m) the samples were retrieved? In L. 139 you wrote "up to 5 CTD depths over the upper 100m", but that is the only information provided.

L. 135-138. Not very clear, be more specific.

L. 153: I think that the cocospheres and coccoliths were identified not only to species level, since morphotypes were also separated.

L. 166: >99%, on average?

L. 170. Delete "and Poulton et al. (2011)" since they followed Young et al (2003) in their paper.

L. 171. "...and central area open or with a thin plate". Based on the morphological study of culture strains by SEM, Hagino et al. (2011) suggested to separate coccoliths with an open central area as Type O from existing morphotypes B, B/ C, and C, characterized by coccoliths with a solid plate in the central area. I wonder why the authors did not separate morphotype O from B/C considering that Type O is extensively distributed in the Southern Ocean (e.g., Hagino et al., 2011; Malinverno et al., 2015).

L. 322: If you mention Pappomonas spp. (L. 324) and Papposphaera spp. (L. 325) you should use "coccolithophore taxa" instead of "coccolithophore species".

L. 322: were identified as coccospheres? or as detached coccoliths? Specify.

L. 326: (Charalampopoulou, 2011). I do not think you need to cite it when she is the first author of this manuscript.

L. 326: "all the way across Drake Passage" might be misleading when looking at Fig.

4.

L. 384-385: Since the section 3.4 refers to morphometric measurements performed on Emiliania huxleyi specimens (see L. 165, section 2.4), you should specify that there, in section 3.4 (e. g. using "Emiliania huxleyi placolith size" instead of just "coccolith size").

L. 400: I could not find Fig. 7!

L. 410 and L. 422: You did not talk about diversity before. I would suggest adding something about diversity in section 3.2.

L. 430: It would be worthwhile to consider Malinverno et al. (2015) and Saavedra-Pellitero et al. (2014) here and/or in L. 55-57.

L. 619: Make clear that this refers to coccolithophore communities in the Iceland Basin/North Hemisphere.

L. 970, 980, 985 and 990: I suggest plotting both transects N-S in Figures 2, 4, 5 and 6 instead of N-S-N. In that way it will be easier for the reader to compare Transect 1 and 2.

Technical corrections:

L. 342: (0.5-1.8 cells mL-1)

L. 347: (<0.01 $\times$ 103 coccoliths-1 mL-1)

L. 374: (0.4 cells mL-1)

L. 497: Winter et al.

L 708: Baumann, K.-H.

L. 762: 257 pp.

L. 791: 401 pp.

L 830: Whitworth III, T.

L 870: Baumann, K.-H.

L. 880: 327 pp.

L 887: pp. 75-97

L 896: Whitworth III, T.

---

## Author Comment (AC1) · 28 Aug 2016

Response to Anonymous Referee #1

This paper is well written and presents new data on coccolithophore distribution and calcite production in the Drake Passage. There is also a lot of accompanying physical and chemical data that are not usually taken on such cruises. Although conclusions are not definitive they will contribute to the ongoing discussion of the role of coccolithophores in the Anthropocene.

Response: We thank the reviewer for their positive comments.

Line 13 Although coccolithophore diversity or abundance?

Response: We actually mean in terms of both numerical abundance and species diversity and have now clarified the opening statement in the abstract to better reflect this.

Line 19 Although CP represents less than 1% of total carbon fixations How does this compare to what has been written about the Great Calcite Belt? How does the unique doubling time of coccolithophores influence the equation?

Response : At the time of this article being submitted to Biogeosciences there was no published information from the Great Calcite Belt, but now Balch et al. (2016, Global Biogeochemical Cycles 30) is available to compare our Drake Passage rates with. Generally, ratios of calcification relative to primary production (photosynthesis) across the Atlantic sector of the Great Calcite Belt (south of ∼50oS) are similar to ratios seen in Drake Passage (<0.02 or ∼2%), although ratios are slightly elevated in the Indian sector of the belt (0.02-0.06 or ∼2-6%). It should be noted that the Great Calcite Belt spans ∼45-55oS and hence is more northerly than most of our Drake Passage (54-64oS) observations. We have now added direct reference to the Balch et al. (2016) study in our revised manuscript, specifically line 550 ('The ratio of CP to primary production for the bulk phytoplankton community (data not shown) was also similar to (sub-) tropical communities (0.001-0.069; Poulton et al., 2007) and lower than those generally in the Great Calcite Belt (Balch et al., 2016)'). We have also added this reference to citations and comments about iron control on coccolithophores (Ln 643: 'More recently, iron has been shown to control the distribution and growth of coccolithophores in the Great Calcite Belt (Balch et al., 2016) in the southern hemisphere') and the role of upwelling/frontal influences (Ln 454 and Ln 457).

Line 71 Length of (what) exposure.

Response: Here we were referring to the length of exposure to high pCO2 (low pH) and we have now clarified this in the revised manuscript (Ln 73).

Line 113 Sampling What is the importance of the temporal timing of the sampling?

Would it make a difference? Jan or March?

Response: Seasonal studies of coccolithophores in the Southern Ocean are extremely limited, and we have only one cruise worth of data. We interpret the reviewers question as 'whether seasonality would have any influence on our conclusions' (see Lns 673-692). Although we can only speculate, seasonal changes in cell numbers (and hence bulk calcification rates) may occur, though the dominant morphotype (B/C) is unlikely to change, and hence cell-specific rates may remain largely unchanged (without significant changes in growth conditions). Hence, our conclusions on B/C morphotype dominance and low cell-specific rates compared to the subpolar North Atlantic are unlikely to change. Our statistical relationships between environmental conditions and coccolithophore dynamics are also unlikely to change seasonally – the relationships primary exist because of the strong latitudinal gradients in temperature, irradiance and saturation state, which may vary seasonally along the survey track, but we would argue that strong latitudinal differences will remain despite such seasonality.

Line 117 samples were collected from the upper 100m of the water column Is it possible to get the depths of the samples? Later line 139 "5 CTD depths over the upper 100 m". Also in Figure 4: line 949 Abundance of Major coccolithophore species Which depths?: : :Not clearly stated the difference between Fig 5 surface distribution of coccolithophores and Fig 4.

Response: We apologise for not making this clearer in the previous draft. Water samples were collected from 5 m, 10 m, 50 m, 75 m and 100 m (note added to Ln 145). Both Figure 4 and 5 include surface water (5 m) abundance and calcification rate data: hence there is no difference between surface species distribution and rate data. We have now altered the Figure legend for Fig. 4 to reflect the sampling depth.

Line 136 as well as the criteria of Orsi et al What criteria?

Response: Orsi et al. (1995) use water mass properties (salinity, temperature, density) to differentiate the different fronts/water masses across the Antarctic Circumpolar Current (ACC). This is now clarified in the text: "...as well as the hydrographic criteria of Orsi et al. (1995)..." (Ln 138).

Line 320 How do the coccolithophores assemblages found in the Drake Passage compare to other publications of coccolithophores in the southern ocean?

Response: We are a little confused with this comment with regards to Ln 320 (now Ln 337 onwards) which is in the Results section - the first section of the Discussion directly addresses this issue (Lns 444 onwards).

Line 322 Drake Passage: including???? "Including" is not a clear word.

Response: We have now deleted 'including' to clarify.

Line 430 A previous study across the Drake Passage When was this study undertaken?

Response: The Holligan et al. (2010) study was carried out in December 2006. We have now added this to Ln 450.

Line 497 Winer et al Winter et al

Response: We thank the reviewer for spotting this mistake and have now corrected it.

Line 528 coccolithophoroes contributed only a small fraction What were the main contributors

Response: Our estimates of a 1% contribution from coccolithophores to total carbon fixation is based on information on the coccolithophores and we can only speculate as to what other phytoplankton groups contributed to the majority of primary production. Diatoms and small flagellates (e.g., Cryptophytes) are likely to be a high proportion of the primary production (see e.g., Sommer and Stabel (1986), Pedros-Alio et al. (1996)).

References Pedrós-Alió C, Calderón-Paz JI, Guixa N, Navarrete A, Vaqué D, Microbial plankton across Drake Passage. Polar Biology 16(8), 613-622 (1996). Sommer U,

Stabel H-H, Near surface nutrient and phytoplankton distribution in the Drake Passage during Early December. Polar Biology 6, 107-110 (1986).

---

## Author Comment (AC2) · 28 Aug 2016

Response to review by M. Saavedra-Pellitero (Referee) msaavedr@uni-bremen.de

The present study by Charalampopoulou et al. addresses the composition and structure of coccolithophore communities and calcite production across Drake Passage (Southern Ocean), regarding also primary production, chlorophyll-a, nutrient concentration, temperature, salinity, irradiance and carbonate chemistry parameters. The manuscript is well written and adds an interesting contribution to the ecology of coccolithophores at high latitudes in a marked environmental N-S gradient. I consider that

this manuscript is novel and addresses compelling scientific questions within the scope of Biogeosciences.

Response: We thank the reviewer for their positive comments.

Specific comments: L. 44: What about morphotype C described in Young et al. (2003)?

Response: Indeed there are other E. huxleyi morphotypes (C, O, T) described in the literature, however this statement refers to the main ones which have recognisable differences in calcite content and quantifiable differences (e.g., differences in distal shield length or element thickness or central area characteristics). Morphotype C appears to be a small version of the B/C coccolith and as the size description in Young et al. (2003) does not clearly differentiate B/C and C we have chosen not to mention it in this case (i.e. how would a study quantifiably and statistically differentiate a mixed population of B/C and C morphotypes?).

L. 97: Would it be possible to add other references here on top of Winter et al. (2014)?

Response: We have now cited Malinverno et al. (2015) here and elsewhere in the discussion to emphasise that not all studies agree with the conclusions of Winter et al. (2014).

L. 123: Section 2.2 Study area should go before section 2.1 Sampling.

Response: We have now swopped these two sections as suggested by the reviewer.

L. 117. Is it possible to know at which depths (0-100m) the samples were retrieved?

Response: We apologise for not making this clearer in the previous draft to either reviewer. Water samples were collected from 5 m, 10 m, 50 m, 75 m and 100 m (not added to Ln 145). Both Figure 4 and 5 include surface water (5 m) abundance and calcification rate data: hence there is no difference between surface species distribution and rate data. We have now altered the Figure legend for Fig. 4 to reflect the sampling depth.

In L. 139 you wrote "up to 5 CTD depths over the upper 100m", but that is the only information provided.

Response: See previous response.

L. 135-138. Not very clear, be more specific.

Response: We have now been more specific about the Orsi et al. (1995) criteria to differentiate the different fronts/water masses across the Antarctic Circumpolar Current (ACC). This is now clarified in the text: "…as well as the hydrographic criteria of Orsi et al. (1995)…".

L. 153: I think that the cocospheres and coccoliths were identified not only to species level, since morphotypes were also separated.

Response: We have corrected this statement to reflect that E. huxleyi was also differentiated into morphotypes based on the criteria of Young et al. (2003) and Poulton et al. (2011).

L. 166: >99%, on average?

Response: Detached coccoliths were predominantly from E. huxleyi in terms of total numbers. We have clarified this in the revised paper.

L. 170. Delete "and Poulton et al. (2011)" since they followed Young et al (2003) in their paper.

Response: We have now deleted Poulton et al. (2011) from this section.

L. 171. ": : :and central area open or with a thin plate". Based on the morphological study of culture strains by SEM, Hagino et al. (2011) suggested to separate coccoliths with an open central area as Type O from existing morphotypes B, B/ C, and C, characterized by coccoliths with a solid plate in the central area. I wonder why the authors did not separate morphotype O from B/C considering that Type O is extensively distributed in the Southern Ocean (e.g., Hagino et al., 2011; Malinverno et al., 2015).

Response: We are aware of the study by Hagino et al. (2011) however made no attempt to differentiate types B/C and O in the present study. This stems from two reasons: (1) we have never been able to resolve the question of whether preservation of the central plate in B/C coccoliths depends on sample processing or degree of calcification; and (2) our focus in this study was coccolith calcite content, where differences between type A and B/C are likely to be significant, whereas differences between type B/C and O are likely to be minimal. We have now changed the wording in this section of the paper to recognise the possible differentiation of B/C and O coccoliths.

L. 322: If you mention Pappomonas spp. (L. 324) and Papposphaera spp. (L. 325) you should use "coccolithophore taxa" instead of "coccolithophore species".

Response: We have now corrected this.

L. 322: were identified as coccospheres? or as detached coccoliths? Specify.

Response: We now specify as coccospheres.

L. 326: (Charalampopoulou, 2011). I do not think you need to cite it when she is the first author of this manuscript.

Response: We have removed this reference to the PhD thesis by the lead author in this instance, but retain it elsewhere as a primary reference to results which are not presented in the manuscript but are contained in the thesis (which is available online).

L. 326: "all the way across Drake Passage" might be misleading when looking at Fig. 4.

Response: Indeed this is slightly misleading and we have removed the 'all the way' from this sentence.

L. 384-385: Since the section 3.4 refers to morphometric measurements performed on Emiliania huxleyi specimens (see L. 165, section 2.4), you should specify that there, in section 3.4 (e. g. using "Emiliania huxleyi placolith size" instead of just "coccolith

size").

Response: We have now retitled this section and added E. huxleyi several times to emphasise that this refers to E. huxleyi only. However, please note that >99% of the coccoliths detected were from E. huxleyi (see Ln 180).

L. 400: I could not find Fig. 7!

Response: We apologise to the reviewer, this should have read Fig. 6A and 6B (Fig. 7 was included in a previous draft).

L. 410 and L. 422: You did not talk about diversity before. I would suggest adding something about diversity in section 3.2.

Response: We have now added species richness data to Figure 5 to highlight the latitudinal trends and show the raw data before the statistical analysis. We have also added the line "Diversity (species richness) generally declined with latitude (Fig. 5b), with the lowest number of species (1-2) present in the Antarctic and Continental Zones of both transects" to the relevant results section in the revised manuscript (Lns 361-363).

L. 430: It would be worthwhile to consider Malinverno et al. (2015) and Saavedra-Pellitero et al. (2014) here and/or in L. 55-57.

Response: We thank the reviewer for pointing out these two key recent publications which we have now added to our references and cited in both the introduction and discussion.

L. 619: Make clear that this refers to coccolithophore communities in the Iceland Basin/North Hemisphere.

Response: We have now been more specific with this line as suggested by the reviewer.

L. 970, 980, 985 and 990: I suggest plotting both transects N-S in Figures 2, 4, 5 and 6

instead of N-S-N. In that way it will be easier for the reader to compare Transect 1 and 2.

Response: We thank the reviewer for the suggestion but believe that the figures and axis are clearly labelled to allow comparison N-S-N. We also hope that this emphasises the decrease in all coccolithophore measurements south of the SB (i.e. mid-plot).

Technical corrections:

L. 342: (0.5-1.8 cells mL-1) Response: Corrected.

L. 347: (<0.01 _ 103 coccoliths-1 mL-1) Response: Corrected.

L. 374: (0.4 cells mL-1) Response: Corrected.

L. 497: Winter et al. Response: Corrected.

L 708: Baumann, K.-H. Response: Corrected.

L. 762: 257 pp. Response: Corrected.

L. 791: 401 pp. Response: Corrected.

L 830: Whitworth III, T. Response: Corrected.

L 870: Baumann, K.-H. Response: Corrected.

L. 880: 327 pp. Response: Corrected.

L 887: pp. 75-97 Response: Corrected.

L 896: Whitworth III, T. Response: Corrected.
* * *

---

## Referee Report (RR1)

Charalampopoulou et al. provided a (new) revised high quality and well written version of the manuscript: "Environmental drivers of coccolithophore abundance and calcification across Drake Passage (Southern Ocean)". As I mentioned in the first review, it is actually a very interesting piece of work.

I went through both reviews (including responses) and checked carefully the new version of the manuscript, and I consider that the authors took into consideration all the suggestions provided. Charalampopoulou et al. extensively replied to the majority of the comments and convincingly rebutted the rest. They even improved by adding updated information from the Great Calcite Belt (Balch et al., 2016), just published last month.

I was a bit confused with the number lines included in the "response to the Anonymous Referee #1", because they do not match with those in the last version of the manuscript (e.g. in the 2$^{nd}$ response line 550 should be 536, 643 should be 631). However, I do not think this is relevant.

I would only comment on:

Line 913: Double check if you need "pp." before the number of pages.
Line 925: The year is missing.
Fig. 5: I would enlarge the dashed line from A to D (instead only to C). That will make it easier for the reader.

---

## Editor Decision (ED1)

Referee #1

Charalampopoulou et al. provided a (new) revised high quality and well written version of the manuscript: "Environmental drivers of coccolithophore abundance and calcification across Drake Passage (Southern Ocean)". As I mentioned in the first review, it is actually a very interesting piece of work.

I went through both reviews (including responses) and checked carefully the new version of the manuscript, and I consider that the authors took into consideration all the suggestions provided. Charalampopoulou et al. extensively replied to the majority of the comments and convincingly rebutted the rest. They even improved by adding updated information from the Great Calcite Belt (Balch et al., 2016), just published last month.

I was a bit confused with the number lines included in the "response to the Anonymous Referee #1", because they do not match with those in the last version of the manuscript (e.g. in the $2^{nd}$ response line 550 should be 536, 643 should be 631). However, I do not think this is relevant.

I would only comment on:

Line 913: Double check if you need "pp." before the number of pages.
Line 925: The year is missing.
Fig. 5: I would enlarge the dashed line from A to D (instead only to C). That will make it easier for the reader.

Editor

Below are a few technical corrections.

Line 147: Please use the math symbol "×", not alphabet "X".

References: Please follow the Copernicus Publications Reference Types described below. For example, please insert "and" between the last author and the second from the last. Journal names are generally abbreviated. Also please use comma, not period, before publication year.

**Copernicus Publications Reference Types**

**General Remarks**

Works cited in my manuscript should be accepted for publication or published already. These references have to be listed **alphabetically** at the end of the manuscript under the **first author's name**. Works "submitted to", "in preparation", "in review", or only available as preprint, should also be included in the reference list. Please do not use bold or italic writing in short citations in the text as well as in the reference list.

Please supply the full author list with last name followed by initials. After the list of authors the complete reference title has to be named. Thereby, journal names are abbreviated according to the ISI Journal Title Abbreviations Index, followed by the volume number, the complete page numbers (first and last page) and the publication year. If the abbreviation of a journal name is not known, please use the full title. In addition to journal articles, all reference types are summarized in the following chapters.

If there is more than one work by the **same first author** his/her papers are listed in the order: (1) single author papers (first author), followed by (2) co-author papers (first author and second author), and finally (3) team papers (first author et al.). Within these three categories the respective papers are then listed as follows:

- **Single author papers**: chronologically, beginning with the oldest. If there is more than one paper in the same year a letter (a, b, c) is added to the year both in the short citation in the text as well as in the reference list.
- **Co-author papers**: first alphabetically according to the second author's last name, and then within each set of co-authors chronologically. If there is more than one paper in the same year per set of co-authors a letter (a, b, c) is added to the year both in the short citation in the text as well as in the reference list.
- **Team papers**: first chronologically (beginning with the oldest), independent of the team author names, then inside one year alphabetically according to the second (third, etc.) author. If there is more than one paper in the same year for a first author (independent of the team) a letter (a, b, c) is added to the year both in the short citation in the text as well as in the reference list.

**Copernicus Publications**
Bahnhofsallee 1e
37081 Göttingen
Germany

Martin Rasmussen (Managing Director)
Dr. Xenia van Edig (Business Development)

**Contact**
publications@copernicus.org
http://publications.copernicus.org
Phone +49-551-900339-50
Fax +49-551-900339-70

**Legal Body**
Copernicus Gesellschaft mbH
Based in Göttingen
Registered in HRB 131 298
County Court Göttingen
Tax Office FA Göttingen
USt-IdNr. DE216566440

[Figure]

In terms of short citations in the text, the ordering can be by relevance, as well as chronologically or alphabetically, depending on author's preference.

**Examples for Reference Sorting**

In general, short citations in the text can be displayed as "[..] Smith (2009) […]", or "[…] (Smith, 2009) […]".

| Reference List | Short Citation |
|---|---|
| **Single author: chronologically** | |
| Smith, P.: …, 2009. | Smith, 2009 |
| Smith, P.: …, 2010a. | Smith, 2010a |
| Smith, P.: …, 2010b. | Smith, 2010b |
| **Co-authors: alphabetically before chronologically** | |
| Smith, P. and Brown, P.: …, 2010. | Smith and Brown, 2010 |
| Smith, P. and Carter, T.: …, 2007. | Smith and Carter, 2007 |
| Smith, P. and Carter, T.: …, 2010a. | Smith and Carter, 2010a |
| Smith, P. and Carter, T.: …, 2010b. | Smith and Carter, 2010b |
| Smith, P. and Thomson, A.: …, 2005. | Smith and Thomson, 2005 |
| **Team: chronologically before alphabetically** | |
| Smith, P., Thomson, A., and Carter, T.: …, 2006. | Smith et al., 2006 |
| Smith, P., Carter, T., and Hanson, M. B.: …, 2008a. | Smith et al., 2008a |
| Smith, P., Carter, T., and Walter, N.: …, 2008b. | Smith et al., 2008b |
| Smith, P., Carter, T., and Hanson, M. B.: …, 2009. | Smith et al., 2009 |
| Smith, P., Brown, P., and Walter, N.: …, 2010. | Smith et al., 2010 |

**Copernicus Publications**
Bahnhofsallee 1e
37081 Göttingen
Germany

Martin Rasmussen (Managing Director)
Dr. Xenia van Edig (Business Development)

**Contact**
publications@copernicus.org
http://publications.copernicus.org
Phone +49-551-900339-50
Fax +49-551-900339-70

**Legal Body**
Copernicus Gesellschaft mbH
Based in Göttingen
Registered in HRB 131 298
County Court Göttingen
Tax Office FA Göttingen
USt-IdNr. DE216566440

**Reference Types & Examples**

**Journal Article**

- Author(s) (Initials always after last name!)
- Article title
- Journal title abbreviation
- Volume
- Page numbers
- Year

Punge, H. J. and Giorgetta, M. A.: Differences between the QBO in the first and in the second half of the ERA-40 reanalysis, Atmos. Chem. Phys., 7, 599–608, 2007.

**Journal Article with doi Number**

- Author(s) (Initials always after last name!)
- Article title
- Journal title abbreviation
- Volume
- Page numbers or article number
- doi number
- Year

Felder, M., Poli, P., and Joiner, J.: Errors induced by ozone field horizontal inhomogeneities into simulated nadir-viewing orbital backscatter UV measurements, J. Geophys. Res., 112, D01303, doi:10.1029/2005JD006769, 2007.

**Book**

- Author(s), Editor(s) (Initials always after last name!)
- Book title
- Edition
- Series title and volume (if any)
- Editors (if not authors)
- Publisher
- Location
- Total pages (optional) pp.
- Year

**Copernicus Publications**
Bahnhofsallee 1e
37081 Göttingen
Germany

Martin Rasmussen (Managing Director)
Dr. Xenia van Edig (Business Development)

**Contact**
publications@copernicus.org
http://publications.copernicus.org
Phone +49-551-900339-50
Fax +49-551-900339-70

**Legal Body**
Copernicus Gesellschaft mbH
Based in Göttingen
Registered in HRB 131 298
County Court Göttingen
Tax Office FA Göttingen
USt-IdNr. DE216566440

[Figure]

Singh, O. N. and Fabian, P. (Eds.): Atmospheric Ozone: a Millennium Issue, Copernicus Publications, Katlenburg-Lindau, Germany, 2003.

**Book Chapter**

- Author(s) (Initials always after last name!)
- Article title
- Book title
- Edition (if any)
- Editors (if any)
- Publisher
- Location
- Page numbers of article in book
- Year

Eagleson, P. S.: Physical composition of the oceans and lakes, in: Dynamic Hydrology, EGU Reprint Series, 2, Copernicus Publications, Katlenburg-Lindau, Germany, 67–78, 2003.

**Presented Paper**

- Author(s) (Initials always after last name!)
- Paper title
- Name of Meeting/Conference
- Location of Meeting/Conference
- Date of Meeting/Conference
- Abstract number
- Year

Keppler, F., Hamilton, J., Braß, M., and Röckmann, T.: An overlooked major source of atmospheric methane: in situ formation in plants, EGU General Assembly, Vienna, Austria, 2–7 April 2006, EGU06-A-08188, 2006.

**Copernicus Publications**
Bahnhofsallee 1e
37081 Göttingen
Germany

Martin Rasmussen (Managing Director)
Dr. Xenia van Edig (Business Development)

**Contact**
publications@copernicus.org
http://publications.copernicus.org
Phone +49-551-900339-50
Fax +49-551-900339-70

**Legal Body**
Copernicus Gesellschaft mbH
Based in Göttingen
Registered in HRB 131 298
County Court Göttingen
Tax Office FA Göttingen
USt-IdNr. DE216566440

[Figure]

**Presented Paper published in Conference Proceedings**

- Author(s) (Initials always after last name!)
- Paper title
- Proceedings title
- Name of Meeting/Conference
- Location of Meeting/Conference
- Date of Meeting/Conference
- Abstract number or page numbers
- Year

Iwata, M., Matsumoto, H., and Kojima, H.: Computer experiments on the plasma wave generation in the vicinity of Earths bow shock, in: Proceedings of the 6th International School/Symposium on Space Plasma Simulation Overview, Garching, Germany, 3–8 September 2001, 4–6, 2001.

**Report, Map, Thesis, Dissertation**

- Author(s) (Initials always after last name!)
- Title
- Report designator (M.S., Ph.D., etc.)
- Issuing Organization/University
- Location
- Total pages (optional) pp.
- Year

Monger, J. W. H. and Journeay, J. M.: Guide to the geology and tectonic evolution of the southern Coast Mountains, Geol. Surv. of Can., Ottawa, Ont., Open File Rep. 2490, 77 pp., 1994.

Brown, R. J. E.: Permafrost in Canada, Geol. Surv. of Can., Ottawa, Ont., Map 1246A, 1967.
Kronberg, E. A.: Dynamics of the Jovian Magnetotail, Ph.D. thesis, International Max Planck Research School, Universities of Braunschweig and Göttingen, Germany, 133 pp., 2006.

**Webpages**

- Title
- URL
- Access date
- Year (if not analog with access date)

**Copernicus Publications**
Bahnhofsallee 1e
37081 Göttingen
Germany

Martin Rasmussen (Managing Director)
Dr. Xenia van Edig (Business Development)

**Contact**
publications@copernicus.org
http://publications.copernicus.org
Phone +49-551-900339-50
Fax +49-551-900339-70

**Legal Body**
Copernicus Gesellschaft mbH
Based in Göttingen
Registered in HRB 131 298
County Court Göttingen
Tax Office FA Göttingen
USt-IdNr. DE216566440

[Figure]

Copernicus Publications: http://publications.copernicus.org/, last access: 2 July 2007.

If an article is available via the internet, an URL address can be inserted before the year, e.g. "available at: http://www.copernicus.org/, 2007."

**Copernicus Publications**
Bahnhofsallee 1e
37081 Göttingen
Germany

Martin Rasmussen (Managing Director)
Dr. Xenia van Edig (Business Development)

**Contact**
publications@copernicus.org
http://publications.copernicus.org
Phone +49-551-900339-50
Fax +49-551-900339-70

**Legal Body**
Copernicus Gesellschaft mbH
Based in Göttingen
Registered in HRB 131 298
County Court Göttingen
Tax Office FA Göttingen
USt-IdNr. DE216566440